# Exergoeconomic Analysis of a Mechanical Compression Refrigeration Unit Run by an ORC

**DOI:** 10.3390/e25111531

**Published:** 2023-11-10

**Authors:** Daniel Taban, Valentin Apostol, Lavinia Grosu, Mugur C. Balan, Horatiu Pop, Catalina Dobre, Alexandru Dobrovicescu

**Affiliations:** 1Department of Engineering Thermodynamics, National University of Science and Technology Politehnica Bucharest, 060042 Bucharest, Romania; tabandaniel@yahoo.com (D.T.); valentin.apostol@magr.ro (V.A.); horatiu.pop@upb.ro (H.P.); 2Lab Energet Mech & Electromagnetism (LEME), University of Paris Nanterre, 50 Rue Sevres, F-92410 Ville d’Avray, France; mgrosu@parisnanterre.fr; 3Department of Thermodynamics, Technical University of Cluj-Napoca, 400114 Cluj-Napoca, Romania; mugur.balan@termo.utcluj.ro; 4Academy of Romanian Scientists, Ilfov 3, 050044 Bucharest, Romania

**Keywords:** exergy analysis, diesel engine, heat exchanger, exergy destruction, exergoeconomic correlation, exergoeconomic cost assessment, exergoeconomic factor

## Abstract

To improve the efficiency of a diesel internal combustion engine (ICE), the waste heat carried out by the combustion gases can be recovered with an organic Rankine cycle (ORC) that further drives a vapor compression refrigeration cycle (VCRC). This work offers an exergoeconomic optimization methodology of the VCRC-ORC group. The exergetic analysis highlights the changes that can be made to the system structure to reduce the exergy destruction associated with internal irreversibilities. Thus, the preheating of the ORC fluid with the help of an internal heat exchanger leads to a decrease in the share of exergy destruction in the ORC boiler by 4.19% and, finally, to an increase in the global exergetic yield by 2.03% and, implicitly, in the COP of the ORC-VCRC installation. Exergoeconomic correlations are built for each individual piece of equipment. The mathematical model for calculating the monetary costs for each flow of substance and energy in the system is presented. Following the evolution of the exergoeconomic performance parameters, the optimization strategy is developed to reduce the exergy consumption in the system by choosing larger or higher-performance equipment. When reducing the temperature differences in the system heat exchangers (ORC boiler, condenser, and VCRC evaporator), the unitary cost of the refrigeration drops by 44%. The increase in the isentropic efficiency of the ORC expander and VCRC compressor further reduces the unitary cost of refrigeration by another 15%. Following the optimization procedure, the cost of the cooling unit drops by half. The cost of diesel fuel has a major influence on the unit cost of cooling. A doubling of the cost of diesel fuel leads to an 80% increase in the cost of the cold unit. The original merit of the work is to present a detailed and comprehensive model of optimization based on exergoeconomic principles that can serve as an example for any thermal system optimization.

## 1. Introduction

The increase in the cost of fossil fuels, demand for energy, and environmental concerns have led to the analysis, design, and development of thermal systems that can also convert low- and medium-temperature-level heat sources into mechanical work. One such thermal system, used on a large scale to recover waste heat, is the Rankine cycle with organic fluids (ORC) [1].

Daniarta, S. et al. [2] analyzed the benefits for using an ORC for recovering the heat released by the baking ovens in an automotive paint shop. The use of the ORC succeeds in reducing the heat loss that accompanies the manufacturing process, contributing to ameliorating the employees’ comfort while also producing electrical energy.

The mechanical power produced by an ORC is often used to drive a VCRC.

Aphornratana, S. and Sriveerakul, T. [3] presented a heat-fed refrigeration cycle concept that combines an ORC and a mechanical vapor compression (VCRC) refrigeration plant using a unitary assembly consisting of a free piston compressor–expander. The two systems use the same working fluid and share the same condenser. Also, the authors of paper [4] presented a thermally activated cooling cycle consisting of an ORC and a VCRC. The system can be powered by solar thermal energy, geothermal energy, or various waste heat flows. The ORC expander shaft was directly coupled to the VCRC compressor shaft to reduce power conversion losses.

Hu, K. et al. [5] focused their attention on the analysis of a refrigerating system based on an ORC. The refrigeration was obtained in a two-stage transcritical CO_2_ mechanical compression cycle. The analysis was based on economic, energetic, and exergetic criteria. The influences of the intermediary and high-pressure CO_2_ were shown. The analysis pointed out that the boiler of the ORC had the largest exergy destruction, while the evaporator of the refrigerator had the highest investment cost.

The advantage of a combined ORC-VCRC compared with absorption refrigeration systems [6] is that when the refrigerating effect is not required, all the thermal energy can be converted into power and used for other applications.

Li, T. et al. [7] analyzed the use of different working fluids in a combined cooling, heating, and power cogeneration system composed of an ORC supplied with geothermal heat and a two-stage mechanical compression refrigerator that offers two levels of cold temperatures. The influence of the geothermal fluid temperature on the turbine net power output was discussed. The values of the mechanical-power-generated cooling and heating when using different working fluids were shown.

Yue, C. et al. [8] proposed that the energy source supplying the secondary systems of a vehicle should be constituted by a combined ORC-VCRC. They used R134a, R245fa, n-propane, and cyclopentane and found R134a to be the most suitable working fluid. The proposed combined system was from 9.2% to 9.8% more efficient than the conventional power system.

Other studies have focused on applying ORC technology to recover waste heat from diesel engines. The authors of paper [9] presented an alternative for using the waste heat from a diesel engine in an ORC, which, in turn, produced the energy needed to drive the compressor of a VCRC that achieved the cooling effect at different temperature levels. 

Ochoa, G. et al. [10] analyzed the use of solar energy as a heat source in ORC cycles.

To increase the overall performance of the combined ORC-VCRC, numerous studies were carried out based on the principles of thermodynamics and economic analysis to determine the operating characteristics of the system, such as the right working fluid, the temperature of the heat source, and the evaporation temperature of the working fluid. 

Bett, A.K. and Jalilinasrabady, S. [11] combined exergy with pinch analysis to optimize an ORC system that recovers geothermal energy. Different ORC fluids were comparatively analyzed. The use of a regenerative internal heat exchanger was considered. The variation in the pinch temperature difference in the ORC evaporator imposed the inlet pressure of the ORC fluid in the turbine. The optimal boiling pressure for the net generated power was found for each ORC fluid.

Camero, A.B. et al. [12] used an advanced exergoenvironmental analysis to discover perspectives for improving an ORC that recovers waste heat from a natural gas engine. The work, based on the concepts of endogenous and exogenous exergy destruction, pointed out that most key parts of the system were weakly connected with the others and could be optimized practically and individually. The analysis revealed that the greatest endogenous exergy destruction happens in the heat transfer process between the engine exhaust gases and ORC fluid, where about 70% of the available energy of the exhaust gases is destroyed. A reduction in this exergy destruction will increase the overall system performance.

Kim and Perez-Blanco, H. [13], looking for the best exergetic efficiency for an ORC-VCRC, analyzed eight fluids, including R143a, R134a, R152a, ammonia, R22, R600, isobutane, and propane. They proposed isobutane as a suitable working fluid to obtain a high coefficient of performance (COP) and an increased exergetic efficiency.

Saleh, B. [14] studied the behavior of an ORC-VCRC for 14 pure working fluids, including hexane, butane, R152a, R245fa, isobutane, RC318, R236ea, R236fa, R1234ze (E), C5F12, RE245cb2, R245ca, R6010, and R6012. From the exergy analysis, the author found that by working with R6012, the ORC-VCRC reached the maximum coefficient of performance. 

Nazari, N. et al. [15] conducted an exergoeconomic analysis and multiobjective optimization for a solar- or biomass-based trigeneration system. In this work, exergy analysis is used only to point out the thermodynamic performance of the system and not as an instrument to find ways for improvement. The economic correlations for calculating the purchase costs of each piece of equipment, except for the compressor, are mainly economic but not exergoeconomic and are based on the size and not on the coefficient of performance of the equipment. The optimization search is based on a multi-verse optimization algorithm and not on exergoeconomic principles.

Wang, Q. et al. [16] analyzed, from the technoeconomic point of view, the performance of an organic Rankine cycle with dual-level heat sources. An exergetic analysis was performed giving only the results of the exergy destructions in each piece of equipment but without any mathematical model to support the analysis.

Louay Elmorsy et al. [17] compared integrated solar combined-cycle configurations based on exergoeconomic evaluation. Five configurations were analyzed. Linear Fresnel and parabolic through collectors and a solar tower were discussed based on exergoeconomic principles. The paper offers detailed economic and exergoeconomic analyses for the best promising configurations. 

Ibrahim et al. [18] analyzed a solar distillation system for cost optimization. The authors used the exergoeconomic method to evaluate the cost parameters. To investigate, for freshwater production, the effects of the key variables on the exergoeconomic cost, a sensitivity analysis was carried out. The analysis was performed on an existing system; and after the exergoeconomic optimization, the cost of the fresh water decreased by about 45%.

Rangel-Hernandez et al. [19] used exergoeconomics to compare the operation of a domestic refrigerator when R1234yf was used as a drop-in replacement for R134a. From the point of view of the unit exergy cost, R134a performed better than R1234yf under different operating conditions due to less exergy destruction in the system.

Fergani, Z. et al. [20] evaluated an organic Rankine cycle with a zeotropic mixture, using exergy-based methods. For parametric optimization, a multiobjective approach was applied. A comparative analysis of zeotropic mixtures and pure fluids was conducted as well. For each piece of equipment, economic and exergoeconomic correlations were given. The conclusion of the work is that the application of zeotropic mixtures as working fluids for ORCs improves the exergoeconomic performances compared to those obtained using the pure components.

Luo, I. et al. [21] evaluated, from thermodynamic and economic viewpoints, the performance of a vapor compression refrigeration machine with CO_2_ as the working fluid driven by a Brayton cycle. The refrigeration was dedicated to hot climate conditions. The objective of the optimization was the maximization of the exegetic efficiency. The optimization search was based on a genetic algorithm and revealed a 4.7% exergetic efficiency when the system operated only as a refrigerator and 22% in cogeneration with heat capacities.

Tashtoush, B. et al. [22] conducted an exergoeconomic analysis of a solar-energy-driven organic Rankine cycle combined with an ejector refrigeration system to produce cooling and electrical power. Exergy and exergoeconomic analyses of the combined system were carried out to predict the cost of the inefficiencies present in the key components of the system.

To further improve the overall performance of ORC-based systems, researchers have used several optimization techniques. Salim, M.S. and Kim, M. [23] presented the application of a genetic algorithm (GA) for the thermoeconomic optimization of the combined ORC-VCRC system to determine the optimal component sizes and thermal performance of the system. The authors recommended R245fa as the working fluid for the optimal thermoeconomic performance of the system. In paper [24], the authors used the TOPSIS technique based on the entropy method to determine the optimal mass fraction of mixtures to be used as working fluids in a simple ORC and concluded that a mass fraction of 0.1/0.9 was optimal for mixtures, such as pentane/butane, hexane/pentane, and isohexane/pentane. Also, in paper [25], Bademlioglu, A.H. et al. presented a multiobjective optimization for the simple ORC system, using the Taguchi technique and gray relational analysis (GRA) to determine the order of importance of each input parameter in the system and their effects on the energetic and exergetic performances of the system. The authors found that by choosing the optimal condensing and evaporation temperatures, the turbine isentropic efficiency and heat-exchanger efficiencies were the most effective, followed by optimizing the superheating temperature, pinch temperature differences in the condenser and evaporator, and isentropic efficiency of the pump.

The literature review reveals the interest in the optimization of the ORC-VCRC compound system. In most approaches, exergy analysis is used for estimating the magnitude of inefficiencies; and for the optimization procedure, researchers appeal to mathematical methods.

In the present work the optimization of an ORC-VCRC will be conducted in an open view based on only exergoeconomic principles. The exergoeconomic optimization offers the possibility to follow the effect of any local structural or operational change on the monetary cost of the overall system and the desired product.

The concept proposed for analysis in this paper combines an ORC with a mechanical vapor compression refrigeration cycle (VCRC) to form a thermally activated cooling system by recovering heat from the combustion gases of a stationary internal combustion engine (ICE). The ORC-VCRC system uses the direct gear ratio of the mechanical work generated by the ORC expander. Thus, the shaft of the ORC expander directly engages the shaft of the VCRC compressor and uses R1224yd(Z) as single working fluid to drive both thermodynamic cycles. In this sense, a hydraulic route has been implemented with a single condenser to circulate the working fluid required for both cycles.

The present work aims to identify and optimize the constructive solutions of the ORC-VCRC-coupled system by means of an exergoeconomic analysis. The exergoeconomic optimization method aims to identify the exergy destruction of each functional area and assign its monetary cost. Areas with a high cost of exergy destruction will be the first targets of the optimization procedures, and the effect of each local cost reduction will be verified at the global level. To quantify the value of a zonal destruction, the fuel, product, loss, and exergetic performance coefficients were evaluated for each operational area, with the aim of obtaining the maximum amount of product from a limited resource at the minimum monetary cost.

The scheme of the ORC-VCRC combined installation and representation of the cycle in the p–h diagram are presented in Figure 1.

The refrigerating cycle (1Tv-6-7-8-1Tv) (Figure 1) is run directly by the ORC (1P-2-3-4-1P) (Figure 1). The refrigerating task of the VCRC is to chill water. When the VCRC works, the mechanical work supplied by the expander of the ORC is totally used to run the VCRC compressor (process 7–8, Figure 1). In the boiler, the ORC recovers the heat carried out by the combustion products of the diesel engine (process 2–3, Figure 1).

The characteristic measures (Table 1) for the analysis of the optimized coupling were taken from the internal combustion engine (ICE) that equips the ICE-ORC experimental stand located within the Department of Thermotechnics, Engines, Thermal Equipment, and Refrigeration at the National University of Science and Technology Politehnica Bucharest. The stand is equipped with a diesel-type engine, model 4TNV98TGGEHR manufactured by Yanmar, and was designed and built to experimentally investigate the possibilities for improving the performance coefficient of an internal combustion engine by coupling it with an ORC installation with the role of recovering the heat dissipated by the engine.

The operating conditions were established starting from the parameters of the heat source for the ORC (engine exhaust gases) and the condenser cooling water. The maximum evaporation temperature of the organic fluid was imposed considering the restriction that the temperature of the combustion gases at the exit from the boiler evaporator should not fall below 140 °C to avoid condensation and the formation of sulfuric acid (H_2_SO_4_) in and implicit corrosion of the ORC boiler.

Compared with absorption- or adsorption-based refrigeration systems, the ORC–VCRC combined system is flexible: During hot summer periods, all the available thermal energy can be converted into mechanical energy and then into a cooling effect; and during winter, when there is no need for cold, the system can produce only electrical energy.

## 2. Mathematical Modeling of the System

The mathematical model is based on mass, energy, and exergy balances for different constructive structures of the ORC-VCRC scheme.

The importance of the exergetic analysis is the possibility to identify and evaluate the magnitude of the dysfunctions in the system. Based on the exergetic analysis, a strategy can be established to improve the performance of a system by making constructive and operational changes.

The analysis of the functional scheme and installation cycle (Figure 1) highlights the following sequence of work processes: (2–20) heating the ORC liquid to the boiling temperature, (20–30) evaporating the ORC liquid in the ORC boiler, (30-3) superheating the vapors in the ORC boiler, (3–4) expanding the vapors to the condensing pressure in the ORC expander, (7–8) compressing the superheated vapors to the condensing pressure in the VCRC compressor, (5) mixing the streams in states (4) and (8), and (5-1) condensing the saturated liquid in the ORC-VCRC condenser. (1TV) is the part of the stream (1) that expands in the throttling valve (TV) of the refrigeration plant. (1P) is the part of the stream (1) that is compressed in the ORC liquid pump (P). (1P-2) is the liquid compression in the ORC pump. (1TV-6) is the throttling of the liquid agent in the VCRC throttling valve. (6–7) is the evaporation and superheating in the VCRC evaporator. (A) is the nodal point for separating the condensed liquid (1) from the streams (1TV) to the TV throttling valve of the VCRC and from (1P) to the ORC liquid pump. (B) is the nodal mixing point of currents (4) and (8).

The analysis was made based on the following assumptions:there is no heat loss in the heat exchangers;the pressure losses in all the components of the installation are neglected;the expansion in the expander, compression in the compressor, and pressure increase in the pump are irreversible adiabatic processes characterized by the isentropic efficiencies ${ɳ}_{sE}$, ${ɳ}_{sCp},$ and ${ɳ}_{sP}$;the expander is directly engaged with the compressor, and power is transferred without mechanical losses;the working fluid in the ORC-VCRC system is R1224yd(Z), a new type of refrigerant that is used to replace R245fa.

The mathematical model of the operation of the ORC-VCRC system is built based on the choice of some decision-making parameters.

### 2.1. Choice of Decision-Making Parameters

#### 2.1.1. The Boiling Temperature in the ORC Boiler

The temperature difference, $∆t_{minB}$, is chosen between the outlet temperature of the combustion gases, $t_{o,g}$, and the boiling temperature of the working fluid in the boiler, $t_{B}$ (Figure 2a). The result is the boiling temperature of the working fluid in the boiler, $t_{B}$.

#### 2.1.2. Condensation Temperature

The temperature difference, $∆t_{minCd}$, is chosen between the outlet temperature of the water from the condenser, $t_{ow,Cd}$, and the condensation temperature of the working fluid, $t_{C}$ (Figure 2b), as follows:(1)$$t_{C}=t_{ow,Cd}+∆t_{minCd}$$

#### 2.1.3. Evaporation Temperature

The temperature difference, $∆t_{minEv}$, is chosen between the outlet temperature of the chilled water from the evaporator, $t_{ow,Ev}$, and the evaporation temperature of the working fluid, $t_{v}$ (Figure 2c), as follows:(2)$$t_{v}=t_{ow,Ev}-∆t_{minEv}$$

#### 2.1.4. The Temperature of the ORC Fluid at the Entrance to the Expander

The degree of superheating of the vapors in the boiler, $∆t_{oh,B}$, is chosen (Figure 1b and Figure 2a), resulting in temperature t_3_ at the exit from the boiler as follows:(3)$$t_{3}=t_{B}+∆t_{oh,B}$$

### 2.2. Energetic Analysis

Based on the energy balances of the component devices, the flow rates of the working fluids, thermal loads of the heat exchangers, and mechanical powers are calculated (Figure 1).

The organic fluid mass-flow rate results from the energy balance of the ORC boiler as follows:(4)$${\dot{Q}}_{B}=\left|{\dot{Q}}_{g}\right|={\dot{m}}_{g}\cdot c_{g}(t_{i,g}-t_{o,g})$$
(5)$${\dot{m}}_{ORC}={\dot{Q}}_{B}/\left(h_{3}-h_{2}\right)$$

The power produced in the expander is as follows:(6)$${\dot{W}}_{E}={\dot{m}}_{ORC}\left(h_{3}-h_{4}\right)$$

The power input of the working fluid drive pump is as follows:(7)$$\left|{\dot{W}}_{P}\right|={\dot{m}}_{ORC}\left(h_{2}-h_{1}\right)$$

The specific mechanical work required to drive the compressor is as follows:(8)$$\left|w_{Cp}\right|=h_{8}-h_{7}$$

Considering that the VCRC compressor is directly driven by the expander without mechanical losses,
(9)$$\left|{\dot{W}}_{Cp}\right|={\dot{W}}_{E}-\left|{\dot{W}}_{P}\right|$$
The mass-flow rate of the fluid in the refrigeration cycle can be calculated as follows:(10)$${\dot{m}}_{VCRC}=\left|{\dot{W}}_{Cp}\right|/\left|w_{Cp}\right|$$

Because the condenser is common to the two combined cycles, its thermal power is as follows:(11)$$\left|{\dot{Q}}_{Cd}\right|=\left({\dot{m}}_{ORC}+{\dot{m}}_{VCRC}\right)\left(h_{5}-h_{1}\right)$$

The energy balance of the evaporator specifies the refrigeration power of the system as follows:(12)$${\dot{Q}}_{Ev}={\dot{m}}_{VCRC}\left(h_{7}-h_{6}\right)$$

The energy performance coefficients are calculated for each individual subsystem and the overall system as follows:(13)$${COP}_{ORC}=\frac{{\dot{W}}_{E}}{\left|{\dot{Q}}_{B}\right|}$$
(14)$${COP}_{VCRC}=\frac{{\dot{Q}}_{ev}}{\left|{\dot{W}}_{cp}\right|}$$
(15)$${COP}_{ORC-VCRC}=\frac{{\dot{Q}}_{ev}}{\left|{\dot{Q}}_{B}\right|}$$

### 2.3. Exergetic Analysis

Exergy represents the maximum mechanical work that a system can release or the minimum mechanical work it must consume to reach total equilibrium with its environment. In energy-transfer processes, a part of the exergy is consumed (destroyed) owing to irreversibility.

Minimizing the exergy destruction in the key components of an energy system provides a strategy to follow to optimize the structure and the way the system works.

The exergy destruction in each operating area of the ORC-VCRC system is calculated based on the Gouy–Stodola equation or the exergy balance equations.

The analysis of the heat transfer in the ORC boiler was carried out by dividing the boiler into distinct areas for liquid heating, evaporation, and vapor superheating as follows:Liquid-heating area (Figure 2a)

The amount of heat required to heat the working fluid in the liquid state in the boiler is as follows:(16)$${\dot{Q}}_{h}={\dot{m}}_{VCRC}\left(h_{20}-h_{2}\right)$$

From the energy balance in this area, the inlet temperature of the combustion gases in this area is as follows:(17)$${\dot{Q}}_{h}={\dot{Q}}_{h,g}={\dot{m}}_{g}c_{g}\left(t_{h,g}-t_{o,g}\right)$$

The exergy destruction associated with heat transfer at the finite temperature difference in the liquid-heating zone is as follows:(18)$${\dot{I}}_{∆{T,}_{h}}=\left|{\dot{E}x}_{Q}^{T_{g,h}}\right|-{\dot{E}x}_{Q}^{T_{ORC,h}}$$
(19)$$\left|{\dot{E}x}_{Q}^{T_{g,h}}\right|=\left|{\dot{Q}}_{g,h}\right|\left(1-\frac{T_{o}}{T_{g,h}}\right)$$
(20)$${\dot{E}x}_{Q}^{T_{ORC,h}}={\dot{m}}_{ORC}\left(h_{20}-h_{2}-T_{o}\left(s_{20}-s_{2}\right)\right)$$
where
(21)$$T_{g,h}=\frac{t_{h,g}-t_{o,g}}{ln\frac{T_{h,g}}{T_{o,g}}}$$

ORC fluid evaporation zone (Figure 2a)

The amount of heat required to evaporate the working fluid in the boiler is as follows:(22)$${\dot{Q}}_{vb}={\dot{m}}_{ORC}\left(h_{30}-h_{20}\right)$$

The exergy of the boiling heat of the ORC fluid in the boiler is as follows:(23)$${\dot{E}x}_{Q}^{T_{ORC,vb}}={\dot{m}}_{ORC}\left(h_{30}-h_{20}-T_{0}\left(s_{30}-s_{20}\right)\right)$$

On the side of the combustion gases in the evaporation area of the ORC fluid, the energy-balance equation specifies the following:(24)$${\left|\dot{Q}\right|}_{g,vb}={\dot{Q}}_{ORC,vb}$$
(25)$${\left|\dot{Q}\right|}_{g,vb}={\dot{m}}_{g}c_{g}\left(t_{vb,g}-t_{h,g}\right)$$
in which the variable is the temperature entering the area of the combustion gases, $t_{vb,g}$.

The average thermodynamic temperature of the combustion gases in the evaporation zone is calculated as follows:(26)$$T_{g,vb}=\left(t_{vb,g}-t_{h,g}\right)/ln\left(T_{vb,g}/T_{h,g}\right)$$

The exergy of the combustion gases is as follows:(27)$$\left|{\dot{E}x}_{Q}^{T_{g,vb}}\right|={\left|\dot{Q}\right|}_{g,vb}\left(1-T_{0}/T_{g,vb}\right)$$

The exergy destruction due to heat transfer at the finite temperature difference for the evaporation of the working fluid is calculated using the following relation:(28)$${\dot{I}}_{∆T,vb}=\left|{\dot{E}x}_{Q}^{T_{g,vb}}\right|-{\dot{E}x}_{Q}^{T_{ORC,vb}}$$

Overheating zone of organic fluid vapors (Figure 2a)

The exergy destruction due to heat transfer at the finite temperature difference for superheating the working fluid is as follows:(29)$${\dot{I}}_{∆T,oh}=\left|{\dot{E}x}_{Q}^{T_{g,oh}}\right|-{\dot{E}x}_{Q}^{T_{ORC,oh}}$$
(30)$${\dot{E}x}_{Q}^{T_{ORC,oh}}={\dot{m}}_{ORC}\left(h_{3}-h_{30}-T_{0}\left(s_{3}-s_{30}\right)\right)$$
(31)$$\left|{\dot{E}x}_{Q}^{T_{g,oh}}\right|={\left|\dot{Q}\right|}_{g,oh}(1-T_{0}/T_{g,oh})$$
(32)$$T_{g,oh}=\left(t_{i,g}-t_{vb,g}\right)/ln\left(T_{i,g}/T_{vb,g}\right)$$

Considering the exergy consumption in the boiler’s functional areas, the total exergy destruction due to heat transfer at the finite temperature difference in the boiler is as follows:(33)$${\dot{I}}_{∆T,B}={\dot{I}}_{∆T,h}+{\dot{I}}_{∆T,vb}+{\dot{I}}_{∆T,oh}$$

Considering Equations (20), (28), and (32), the total exergy consumption in the ORC boiler, which is also the exergetic fuel of the global system, is as follows:(34)$$\left|{\dot{Ex}}_{Q,g}\right|=\left|{\dot{E}x}_{Q}^{T_{g,h}}\right|+\left|{\dot{E}x}_{Q}^{T_{g,vb}}\right|+\left|{\dot{E}x}_{Q}^{T_{g,oh}}\right|$$

For the other equipment components of the ORC-VCRC group (Figure 1), the following exergy consumptions or losses are identified:exergy destruction in the expander;
(35)$${\dot{I}}_{E}={\dot{m}}_{ORC}T_{0}\left(s_{4}-s_{3}\right)$$

exergy loss in the condenser;


(36)
$${\dot{L}}_{Cd}={\dot{m}}_{ORC-VCRC}\left[h_{5}-h_{1}-T_{0}\left(s_{5}-s_{1}\right)\right]$$


exergy destruction due to the mixing of the working fluid at the entrance to the common condenser (state 5, Figure 1);


(37)
$${\dot{I}}_{m}=T_{0}\left({\dot{m}}_{ORC-VCRC}\cdot s_{5}-{\dot{m}}_{ORc}\cdot s_{4}-{\dot{m}}_{VCRC}\cdot s_{8}\right)$$


exergy destruction in the pump;


(38)
$${\dot{I}}_{P}=T_{0}\cdot {\dot{m}}_{ORC}\left(s_{2}-s_{1}\right)$$


exergy destruction in the throttling valve;


(39)
$${\dot{I}}_{TV}=T_{0}\cdot {\dot{m}}_{VCRC}\left(s_{6}-s_{1}\right)$$


exergy destruction in the compressor.


(40)
$${\dot{I}}_{Cp}=T_{0}\cdot {\dot{m}}_{VCRC}\left(s_{8}-s_{7}\right)$$


The product of the combined ORC-VCRC system is the exergy of the refrigerating effect achieved in the evaporator of the refrigerating cycle, as follows:(41)$$\left|{\dot{E}x}_{Q}^{T_{VCRC,v}}\right|=\left|\dot{E}x_{Q_{0}}\right|=-{\dot{m}}_{VCRC}\left(h_{7}-h_{6}-T_{0}\left(s_{7}-s_{6}\right)\right)$$

The exergetic performance of the combined ORC-VCRC system is specified by the global exergetic efficiency and share of the exergy consumption and losses, associated with each equipment or process, in the exergy consumption of the system.

Noting that the product of the global system is defined by Equation (41) and that the fuel used upon entering the system is defined by Equation (34), the exergetic efficiency of the ORC-VCRC system is as follows:(42)$${ɳ}_{ex}=\frac{\left|{\dot{E}x}_{Q}^{T_{VCRC,v}}\right|}{\left|{\dot{Ex}}_{Q,g}\right|}=\frac{\left|\dot{E}x_{Q_{0}}\right|}{\left|{\dot{Ex}}_{Q,g}\right|}$$

The weight of the destruction or loss of exergy in the exergy consumption of the global system is defined as follows:(43)$$\psi_{i}=\frac{{\dot{I}}_{i}}{\left|{\dot{Ex}}_{Q,g}\right|}$$

### 2.4. Results and Discussion for the Basic ORC-VCRC Cycle

For the basic ORC-VCRC scheme (Figure 1), the energetic and exergetic studies were carried out for the fluid R1224yd(Z) under the following conditions: $t_{0}=15\;{}^{\circ}C$; ${\dot{m}}_{g}=0.0534\;kg/s$; $t_{i,g}=480\;{}^{\circ}C$; $t_{o,g}=140\;{}^{\circ}C$; $c_{g}=1.14\;kJ/(kgK)$; $t_{ow,Cd}=22\;{}^{\circ}C$; $t_{v}=-5\;{}^{\circ}C$; ${∆T}_{minB}=10\;K$; ${∆T}_{oh,B}=6\;K$; ${∆T}_{minCd}=6\;K$; ${ɳ}_{sE}=0.85$; ${ɳ}_{sCp}=0.85$; ${ɳ}_{sP}=0.6$.

The results of the energetic and exergetic analyses of the ORC-VCRC-coupled cycle are presented in Table 2 and Figure 3.

The exergy analysis highlights the high exergy consumption due to the heat transfer at the finite temperature difference from the ORC boiler. The largest temperature differences, which also determine the highest exergy consumption, are found in the ORC liquid heating zone, $\psi_{∆T,h}=28.08\%$, and evaporation zone, $\psi_{∆T,vb}=20.96\%$; a reduction in these destructions is the way forward in the optimization procedure. A decrease in the temperature difference in the ORC boiler would lead to a decrease in the exergy destruction in this area, leading to an increase in the cooling power of the combined system. The loss of exergy in the condenser is more related to the temperature difference between the ORC and VCRC fluids and the cooling water, which is influenced to some extent by the variation in the organic fluid mass-flow rates with respect to the variation in the other decision-making parameters.

To understand how the decision-making parameters functionally and constructively influence the evolution of the ORC-VCRC system, a sensitivity study of the energetic and exergetic performances and exergy destruction and losses of the combined system was conducted by varying the decision-making parameters (${∆t}_{minB}$, ${∆t}_{minCd}$, and ${∆t}_{oh,B}$).

#### 2.4.1. The Behavior of the ORC-VCRC Cycle Obtained by Varying the Temperature Difference, ${∆t}_{minB}$, in the ORC Boiler

The results of the sensitivity study of the change in the temperature difference, ${∆t}_{minB}$, in the ORC boiler are presented in Figure 3.

When the temperature difference, $∆t_{minB}$, increases, the exergy destruction due to the heating process, $\psi_{∆T,h}$, decreases and that due to the evaporation process, $\psi_{∆T,vb}$, increases (Figure 3b). Overall, the exergy destruction in the boiler increases with increasing temperature difference between the fluids, which leads to a decrease in the exergetic efficiency of the cycle and a decrease in the exergy of the refrigerating power (Figure 3a).

#### 2.4.2. The Behavior of the ORC-VCRC Cycle Obtained by Varying the Overheating Degree, $∆t_{oh,B}$, in the ORC Boiler

Increasing the degree of overheating in the boiler has a positive effect, leading to an increase in the exergetic efficiency of the combined cycle and an increase in the exergy of the refrigerating power (Figure 4a).

Under the conditions of the constant thermal flow transferred to the ORC boiler from the combustion gases, the increase in the degree of overheating, $∆t_{oh,B}$, of the ORC fluid leads to the redistribution of thermal loads from the boiler in the liquid-heating, -evaporation, and -superheating phases. The relative exergy destructions from the phases of liquid heating, $\psi_{∆T,h}$, and evaporation, $\psi_{∆T,vb}$, which have the highest weights, decrease faster than the increase in the weight of the exergy destruction associated with overheating, $\psi_{∆T,oh}$ (Figure 4b); overall, the exergy destruction caused by the irreversibility of the heat transfer at the finite temperature difference in the boiler, $\psi_{∆T,B}$, decreases, leading to an increase in the exergy efficiency of the global system and an increase in the exergy of the refrigerating power (Figure 4a).

#### 2.4.3. The Behavior of the ORC-VCRC Cycle Obtained by Varying the Minimum Temperature Difference, $∆t_{minCD}$, in the Condenser

When the minimum temperature difference, $∆t_{minCD}$, in the condenser increases, the exergetic efficiency of the ORC-VCRC system and the exergy of the refrigerating power decrease (Figure 5).

The increase in the minimum temperature difference, $∆t_{minCD}$, in the condenser leads to an increase in the condensation temperature, which leads to a decrease in the refrigerating power and its exergy, $\dot{E}x_{Q_{0}}$ (Figure 5a). The decrease in the exergy of the refrigeration power is the factor that determines the decrease in the exergy efficiency of the global system despite the reduction in the relative exergy destruction, $\psi_{∆T,h}$, associated with the heating of the ORC fluid in the boiler and the consequent reduction in the total exergy destruction in the boiler, $\psi_{∆T,B}$ (Figure 5b).

Figure 6 presents, in $\left(\Theta =1-\frac{T_{o}}{T}\right)$ coordinates, the exergetic temperature factor—H—curve of the hot stream represented by the combustion gases and the composite curve of the cold stream represented by the ORC fluid from the ORC boiler.

The area between the flue-gas curve and H axis gives the measure of the total exergy of the heat transferred from the flue gases, and the area between the composite curve of the working fluid (ORC), in the heating, evaporating, and superheating stages, and the H axis gives the measure of the exergy received by the working fluid. The areas between the two curves give the measure of the exergy destruction in each distinct system component, i.e., heater, evaporator, and superheater.

Figure 6a shows that for the simple ORC system (Figure 1), a large part of the amount of exergy introduced through the combustion gases is consumed in the boiler as follows: 28.07% in the heater, 20.96% in the evaporator, and only 2.425% in the superheater. The recorded values are due to the large temperature difference in the different areas of the boiler.

From Figure 6a, it becomes evident that the largest area between the two curves (and the most exergy destruction) is associated with heat transfer at a too-large temperature difference in the heating and evaporating stages of the working fluid. A reduction in these areas and, therefore, the amount of exergy consumed, can be achieved for the same evaporation temperature by increasing the inlet temperature of the ORC fluid in the heating zone; this can be accomplished, if the ORC fluid allows (i.e., if it is a dry type), by changing the structure of the ORC cycle by introducing an internal recuperative exchanger (Figure 7). The greatest effect for reducing the amount of exergy destruction in the boiler is the increase in the evaporation temperature to the limit allowed by the temperature difference at pinch in this device.

#### 2.4.4. ORC-VCRC Scheme with an Internal Heat Exchanger on the ORC Circuit

Figure 7 shows the scheme of the ORC-VCRC with an internal heat exchanger on the ORC circuit.

Table 3 presents the results of the mathematical modeling based on the exergetic analysis of the operation of the ORC-VCRC cycle with an internal heat exchanger on the ORC circuit for a 100% engine load.

By analyzing the results shown in Figure 6a compared with those shown in Figure 6b, the area between the two curves decreased when the internal heat exchanger was introduced to the ORC scheme, especially in the heating and evaporating stages of the working fluid. This decrease in the surface area leads to a decrease in the share of the exergy destruction in the ORC boiler by 4.19% and, finally, an increase in the overall exergetic efficiency by 2.03% and, implicitly, in the COP of the ORC-VCRC installation with the internal heat exchanger.

## 3. Exergoeconomic Optimization of the Basic Scheme of the ORC-VCRC Coupled System

### 3.1. Exergoeconomic Analysis: General Principles

Exergoeconomic analysis is the only investigation method and optimization procedure that takes into account the fact that any thermodynamic system interacts with the following two environments:(a)a physical environment determined by a system of intensive parameters, such as pressure, temperature, and chemical potential;(b)an economic environment characterized by prices of raw materials and equipment and sets of regulations to ensure sustainable development.

In defining the method for searching for optimal functional and constructive solutions, the exergoeconomic analysis is based on the union between the thermodynamics of irreversible processes and the economic analysis.

Given that energy is a conserved measure (principle I of thermodynamics) and that, in principle, it is not consumed and, therefore, cannot conceptually appear in economic balance sheets, another non-conserved measure, i.e., exergy, must be found to define the consumption of usable energy.

The exergoeconomic analysis lays the foundation for establishing a methodology for directly searching for the optimum system constructively and functionally, while offering users clear rules for improving the studied system, for which effects can be followed step by step.

The exergetic analysis method makes the connection between the system and its surrounding physical environment with which it interacts, using the concept of exergy, which quantifies the value of the use of each exergy stream, highlighting the place and size of the consumed (destroyed) exergy.

Written in an economic tone, the exergetic balance equation (Figure 8a)
(44)$${\Sigma Ex}_{i}={\Sigma Ex}_{o}+\Sigma I$$
becomes
(45)$${\dot{E}x}_{F}={\dot{E}x}_{P}+\Sigma {\dot{E}x}_{L}+\Sigma \dot{I}$$
where ${\dot{E}x}_{F}$—the exergy current of the resources used (generically called fuel);${\dot{E}x}_{P}$—the exergy current of the product;${\dot{E}x}_{L}$—the exergy of the loss currents;$\dot{I}$—consumption or destruction of exergy due to irreversibility (of work processes).

**Figure 8 entropy-25-01531-f008:**
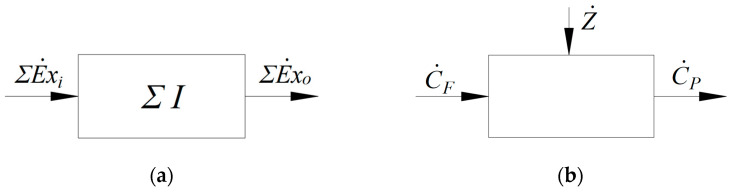
Balance sheets: (**a**) exergy balance for a control volume; (**b**) exergoeconomic balance.

The exergoeconomic analysis balances the economics accounting for both the monetary flows related to the operating process and those of investments (Figure 8b).

The monetary-balance equation is of the following form:(46)$${\dot{C}}_{P}={\dot{C}}_{F}+\dot{Z}$$
where $\dot{C_{P}}\left[\frac{EUR}{s}\right]$—the monetary cost of the product stream;${\dot{C}}_{F}\left[\frac{EUR}{s}\right]$—the monetary cost of the fuel flow;$\dot{Z}\left[\frac{EUR}{s}\right]$—the amortization rate of the invested capital.

For the calculation of the amortization rate of the invested capital, a capital recovery factor, CRF, with the following formula is applied:(47)$$CRF=\left[1+\frac{{\left(1+i_{ef}\right)}^{n}}{{\left(1+i_{ef}\right)}^{n-1}}\right]$$
where $i_{ef}=0.05$ represents the interest (5% per year), and n = 10 represents the economic life period.

The amortization rate of the capital invested for equipment is calculated as follows:(48)$$\dot{Z}\left[\frac{EUR}{s}\right]=Z[EUR]\cdot \frac{CRF}{n_{h}\cdot 3600}$$
where $n_{h}=7000$ h is the number of operating hours per year, and the amortization rate of the invested capital per hour is given as follows:(49)$${\dot{Z}}_{h}\left[\frac{EUR}{hour}\right]=Z\left[\frac{EUR}{s}\right]\cdot 3600[s]$$

If the amount of exergy losses can be reduced by sealing, isolation, or recovery, a reduction in the amount of exergy destruction, which is the essential source for increasing the performance coefficient, is achieved by increasing the investment expenses. The antagonistic evolution of the operating cost (caused by exergy destruction) and investment cost leads to the optimal solution given by the minimization of the total cost as the sum of the operating and investment costs.

The exergoeconomic optimization method aims to identify the exergy destruction of each functional area and assign its monetary cost.

Areas with high exergy-destruction costs will be the first targets of the optimization procedures, and the effect of each local cost reduction will be verified at the global level.

To quantify the value of a zonal destruction, the fuel, product, loss, and exergetic performance coefficients will be evaluated for each operational area.

We will proceed with the search for the functionally and constructively optimal model for the ORC-VCRC combined system in which the power Rankine cycle is simple and does not have an internal recuperative heat exchanger.

The scheme of the VCRC installation operated with the ORC system shown in Figure 9 was divided into the following seven operating areas: (1) ORC Evaporator (Boiler); (2) Expander; (3) Compressor; (4) Condenser; (5) Throttling Valve; (6) VCRC Evaporator; (7) ORC Liquid Pump, to which node A for splitting and node B for mixing of substance currents are added.

The exergetic concepts of Fuel, Product, Exergy Loss, and Destruction for each functional area are presented in Table 4 [26].

Observing from relationship (46) that the destruction, I, of a zonal exergy is a part of the exergy of the fuel, it is logical to assign it as the unitary monetary cost, which is the unitary cost of the fuel in that specific area.

### 3.2. Exergoeconomic Monetary Cost Assessment

Each internal unitary monetary cost represents a part of the overall system’s fuel cost and capital investment. These costs are determined by the system’s interaction with the market.

A methodology must be found to allocate these costs imposed by the market to each internal current.

There are 19 currents: 1–11, 1P, 1TV, α, β, γ, δ, τ, and ξ, for which 19 relationships are needed between them.

Nine monetary-balance relationships of the Equation (46) type can be written for the seven established functional areas and for the two nodes A and B.

The rest of the equations, up to 19, will represent auxiliary relationships between the monetary costs of the various exergy currents.

The auxiliary equations are built based on some rules that can be stated based on the following principles of economic common sense [26]:
*Rule 1* → For an unconsumed fuel stream exiting a subsystem, the monetary cost per unit of exergy is equal to the unit cost of the fuel stream feeding the subsystem;*Rule 2* → If a subsystem has several products, the monetary cost per unit of exergy of each product is the same;*Rule 3* → If a component of a product has several output currents, each is assigned the same unit monetary cost;*Rule 4* → In the absence of an external evaluation, the loss streams are assigned a zero unitary monetary cost;*Rule 5* → In the absence of an external assessment, the unitary exergy costs of the input currents in the global system are equal to their exergies.

A mathematical model based on the exergoeconomic analysis is written for each functional zone (Figure 9) as follows:Zone 1. The ORC evaporator (boiler)

The exergoeconomic monetary-balance cost (Equation (46)) of the ORC boiler is as follows:(50)$$c_{\alpha }\cdot {\dot{E}x}_{\alpha }+c_{2}\cdot {\dot{E}x}_{2}-c_{\beta }\cdot {\dot{E}x}_{\beta }-c_{3}\cdot {\dot{E}x}_{3}+{\dot{Z}}_{B}=0$$

The exergy of the combustion gases, ${\dot{E}x}_{\alpha }$, is calculated as the sum of their thermomechanical and chemical exergies, at a certain operating mode of the engine, compared to the standard composition of the ambient environment.

Because combustion gases are the unconsumed part of diesel fuel (internal combustion engine fuel), they will be assigned the diesel fuel unit exergetic monetary cost (*Rule 1*). It is considered that the mass exergy of diesel is ${\dot{E}x}_{CHF}=41,800$ (kJ/kg) and has a cost of 1.7 (EUR/kg) as follows:(51)$$c_{F}=c_{\alpha }=4.07\cdot 10^{-5}\left[\frac{EUR}{kJ}\right]$$

Because the exergy of current β (Figure 9) is a loss (Table 4), according to *Rule 4*,
(52)$$c_{\beta }=0$$

Zone 2. The expander

The exergoeconomic monetary-balance cost (Equation (46)) of the expander is as follows:(53)$$c_{3}\cdot {\dot{Ex}}_{3}-c_{9}\cdot {\dot{W}}_{E}-c_{4}\cdot {\dot{Ex}}_{4}+{\dot{Z}}_{E}=0$$

Because current 4 represents an unconsumed part of fuel 3, as per *Rule 1*,
(54)$$c_{3}=c_{4}$$

The unit monetary costs of currents 9–11 are equal because they represent parts of the same product (*Rule 3*) as follows:(55)$$c_{10}=c_{9}$$
(56)$$c_{11}=c_{9}$$

Zone 3. The compressor

The exergoeconomic monetary-balance cost (Equation (46)) of the compressor is as follows:(57)$$c_{7}\cdot {\dot{E}x}_{7}+c_{10}\cdot \left|{\dot{W}}_{Cp}\right|-c_{8}\cdot {\dot{E}x}_{8}+{\dot{Z}}_{Cp}=0$$

Zone B. The mixing point

At the mixing point, B, the following monetary-balance equation can be written:(58)$$c_{4}\cdot {\dot{E}x}_{4}+c_{8}\cdot {\dot{E}x}_{8}=c_{5}\cdot {\dot{E}x}_{5}$$

Zone 4. The condenser

The condenser of the ORC-VCRC system is a dissipative zone and has no exergetic product (Table 4).

Its monetary-balance equation, based on exergoeconomic criteria, is as follows:(59)$$c_{5}\cdot {\dot{E}x}_{5}+c_{\xi }\cdot {\dot{E}x}_{\xi }-c_{1}\cdot {\dot{E}x}_{1}-c_{\tau }\cdot {\dot{E}x}_{\tau }+{\dot{Z}}_{Cd}=0$$

Because the exergy of the cooling water of the condenser that is not being used represents a loss, according to *Rule 4*,
(60)$$c_{\xi }=0$$
(61)$$c_{\tau }=0$$

Zone A. The splitting point

The following monetary-balance equation can be written for the splitting point, A:(62)$$c_{1}\cdot {\dot{E}x}_{1}+c_{1TV}\cdot {\dot{E}x}_{1TV}-c_{1P}\cdot {\dot{E}x}_{1P}=0$$
where the specific monetary cost of currents 1TV and 1P is the same as those representing multiple products (*Rule 2*) as follows:(63)$$c_{1TV}=c_{1P}$$

Zone 5. The throttling valve

The exergoeconomic monetary-balance cost (Equation (46)) of the throttling valve is as follows:(64)$$c_{1TV}\cdot {\dot{E}x}_{1TV}-c_{6}\cdot {\dot{E}x}_{6}+{\dot{Z}}_{TV}=0$$

Zone 6. The VCRC evaporator

The exergoeconomic monetary-balance cost (Equation (46)) of the evaporator is as follows:(65)$$c_{6}\cdot {\dot{E}x}_{6}+c_{\gamma }\cdot {\dot{E}x}_{\gamma }-c_{7}\cdot {\dot{E}x}_{7}-c_{\delta }\cdot {\dot{E}x}_{\delta }+{\dot{Z}}_{Ev}=0$$

The exergy current, 7, is the unconsumed part of the fuel, 6, that feeds the evaporator. According to *Rule 1*, the two currents have the same monetary unit cost as follows:(66)$$c_{7}=c_{6}$$

According to *Rule 5*, a value is assigned to current γ as follows:(67)$$c_{\gamma }=1$$

It is observed that the value of the unitary monetary cost of the chilled water current, γ, is not important in the calculation because the state, γ, is taken as a reference for the calculation of the exergy of the chilled water (${\dot{E}x}_{\gamma }=0)$.

Zone 7. The ORC liquid pump

For the ORC liquid pump, the monetary-cost balance equation is as follows:(68)$$c_{1P}\cdot {\dot{E}x}_{1P}+c_{11}\cdot \left|\dot{W_{P}}\right|-c_{2}\cdot {\dot{E}x}_{2}+{\dot{Z}}_{P}=0$$

Equations (50)–(68) represent the 19 equations of the mathematical model for calculating the 19 monetary-unit costs.

### 3.3. Exergoeconomic Cost Correlations of Component Equipment

Unlike thermoeconomic cost correlations, in which the purchase cost of equipment is estimated as a function of the amount of material resources used, exergoeconomic correlations specify the cost according to the exergetic performance coefficient (or defined based on the second law of thermodynamics) or the decision-making variables that define the exergetic performance (such as exergy destruction) and function of the size of the exergetic product of the equipment [27,28,29,30,31,32].

(a)VCRC compressor

The exergoeconomic cost correlation is of the following form [27]:(69)$$Z_{Cp}=71\frac{{\dot{m}}_{VCRC}}{0.9-\eta_{sCp}}{\cdot r}_{Cp}\cdot \ln r_{Cp}\left[EUR\right]$$
where the performance coefficient of the equipment defined based on the second law of thermodynamics is the isentropic efficiency, $\eta_{sCp}$, of the compression process, and the exergetic product is as follows:(70)$${P_{ex})}_{Cp}={\dot{m}}_{VCRC}\cdot r_{Cp}\cdot \ln r_{Cp}\left[\frac{kg}{s}\right]$$
which indicates the antagonistic effect of the variation in the compressed-gas flow rate ${\dot{m}}_{VCRC}$ with the compression ratio, $r_{Cp}=\frac{p_{c}}{p_{v}}=\frac{p_{8}}{p_{7}}$ (Figure 9).

The investment cost, $Z_{Cp}\left[EUR\right]$, has been updated to the level of 2020.

(b)ORC expander

The exergoeconomic correlation for the acquisition cost of the expander (Zone 2, Figure 9) takes into account the isentropic efficiency of the expansion, $\eta_{sE}$, as a measure of the perfection of the equipment through the prism of the second law of thermodynamics. The cost of the expander is proportional to the exergetic product as a measure of the mechanical work produced by expansion as follows [27]:(71)$${P_{ex})}_{D}={\dot{m}}_{ORC}\cdot \ln\frac{p_{3}}{p_{4}}\left[\frac{kg}{s}\right]$$

These notations correspond to those in Figure 9, which shows the technological scheme of the installation.

With these considerations and accounting for the cost update at the level of 2020, the exergoeconomic cost correlation for the expander is as follows [31]:(72)$$Z_{E}=479\frac{{\dot{m}}_{ORC}}{0.92-\eta_{sE}}\cdot \ln\frac{p_{3}}{p_{4}}\left[EUR\right]$$

(c)ORC pump

For the pump, a form of exergoeconomic correlation like that used for the compressor is chosen as follows [28]:(73)$$Z_{P}=10.38\frac{{\dot{m}}_{ORC}}{0.9-\eta_{sP}}r_{P}\cdot \ln r_{P}\left[EUR\right]$$
in which the performance of the equipment in relation to the second law of thermodynamics is described by the isentropic compression efficiency, $\eta_{sP}$, and the exergetic product correlates the pumped flow rate, ${\dot{m}}_{ORC}$, with the pressure increase ratio, $r_{P}=\frac{p_{2}}{p_{1}}$ (Figure 9).

(d)Heat exchangers

In the case of the heat exchangers, to specify the cost of the purchase, the starting point is the following thermoeconomic correlation:(74)$$Z=c\cdot A\left[EUR\right]$$
where $c\left[EUR/m^{2}\right]$ represents the cost of the heat-exchange surface unit; $A\left[m^{2}\right],$ its surface.

To highlight the decision-making parameter that imposes the exergetic performance of the device, the heat-exchange surface is explained according to the minimum temperature difference in the device. The minimum temperature difference, $∆T_{min}$, determines the amount of exergy destruction in the device. This decision-making parameter balances the operating expense (exergy consumption) and acquisition cost (heat-exchange surface) of the device as follows:(75)$$A=\frac{\dot{Q}}{U\cdot ∆T_{m}}=\frac{\dot{m}\cdot ∆h}{U\cdot ∆T_{m}}$$
where $U\;\left[W/(m^{2}\cdot K)\right]$ represents the global heat-exchange coefficient of the device, and $∆T_{m}$ is the average logarithmic temperature difference in the device.

d.1The refrigeration unit evaporator

The temperature variation is plotted as a function of the heat-transfer surface in Figure 2c as follows:(76)$$∆T_{min}^{V}=t_{\delta }-t_{7}$$
(77)$${\Delta T}_{m}^{V}=\frac{t_{\gamma }-t_{\delta }}{\ln\frac{t_{\gamma }-t_{V}}{t_{\delta }-t_{V}}}=\frac{t_{\gamma }-t_{\delta }}{\ln\frac{t_{\gamma }-{(t}_{\delta }-∆T_{min}^{V})}{∆T_{min}^{V}}}=\frac{t_{\gamma }-t_{\delta }}{\ln\frac{t_{\gamma }-t_{\delta }+∆T_{min}^{V}}{∆T_{min}^{V}}}$$
Considering the temperature drop on the cooled fluid side as imposed design data,
(78)$$t_{\gamma }-t_{\delta }=a$$
relationship (77) can be written as follows:(79)$${\Delta T}_{m}^{V}=\frac{a}{\ln\frac{a+∆T_{min}^{V}}{∆T_{min}^{V}}}$$

Taking (79) into account, Equation (74) is as follows:(80)$$Z_{V}=c_{V}\frac{{\dot{Q}}_{V}\ln\frac{a+∆T_{min}^{V}}{∆T_{min}^{V}}}{a\cdot U_{V}}$$

Considering $c_{V}=87\;\left[EUR/m^{2}\right]$ and $U_{V}=500\;\left[W/(m^{2}\cdot K)\right]$ and choosing a = 16 K, Equation (80) can be written as follows:(81)$$Z_{V}\left[EUR\right]=10.87\cdot {\dot{Q}}_{V}\left[kW\right]\ln\left(16\cdot {\left({\Delta T}_{min}^{V}\right)}^{-1}+1\right)$$

d.2Condenser

By applying to the condenser the same scheme used in the case of the VCRC evaporator, the following exergoeconomic correlation is obtained:(82)$$Z_{Cd}=c_{cd}\frac{{\dot{Q}}_{cd}\ln\frac{b+∆T_{min}^{Cd}}{∆T_{min}^{Cd}}}{b\cdot U_{Cd}}$$
where from Figure 2b
(83)$$b=t_{\tau }-t_{\xi }$$
is imposed by the design and
(84)$$∆T_{min}^{Cd}=t_{c}-t_{\tau }$$

Considering that $c_{cd}=254\left[EUR/m^{2}\right]$, $U_{cd}=1750\left[W/(m^{2}\cdot K)\right]$, and b = 7 K, Equation (82) can be written as follows:(85)$$Z_{Cd}\left[EUR\right]=20.73\cdot {\dot{Q}}_{Cd}\left[kW\right]\ln\left(7\cdot {\left({\Delta T}_{min}^{Cd}\right)}^{-1}+1\right)$$

d.3ORC boiler

In the case of the ORC boiler, the exergoeconomic correlation for calculating the purchase cost is of the following form:(86)$$Z_{B}\left[EUR\right]=c_{B}\left[\frac{EUR}{m^{2}}\right]\cdot A_{B}\left[m^{2}\right]$$
where the heat-transfer surface is the sum of the surfaces corresponding to the processes of heating the subcooled liquid ($A_{h})$, boiling ($A_{vb}$), and overheating ($A_{oh})$ as follows:(87)$$A_{B}=A_{h}+A_{vb}+A_{oh}$$

The heat-exchange surfaces of each functional area of the boiler are calculated as follows:

Liquid heating area

(88)$$A_{h}=\frac{{\dot{Q}}_{h}}{U_{h}\cdot ∆T_{m}^{h}}$$
where from Figure 2a
(89)$${\dot{Q}}_{h}={\dot{m}}_{ORC}(h_{20}-h_{2})$$
$U_{h}=70\;\left[W/(m^{2}\cdot K)\right]$ is the global heat-transfer coefficient for the liquid heating process as follows:(90)$$∆T_{h}=\frac{\left(t_{\beta }-t_{2}\right)-(t_{h,g}-t_{20})}{ln\frac{\left(t_{\beta }-t_{2}\right)}{\left(t_{h,g}-t_{20}\right)}}$$

The temperature, $t_{h,g}$, on the heating-gas side results from the energy balance of the liquid-heating zone (Figure 2a) as follows:(91)$$t_{h,g}=t_{\beta }+\frac{{\dot{m}}_{ORC}(h_{20}-h_{2})}{{\dot{m}}_{g}\cdot c_{g}}$$
where $t_{\beta }=140\;{}^{\circ}C$.

Boiling area

(92)$$A_{vb}=\frac{{\dot{Q}}_{vb}}{U_{vb}\cdot ∆T_{m}^{vb}}=\frac{{\dot{m}}_{ORC}(h_{30}-h_{20})}{U_{vb}\cdot ∆T_{m}^{vb}}$$
where $U_{vb}=90\left[W/(m^{2}\cdot K)\right]$ is the global heat-transfer coefficient for the boiling process as follows:(93)$$∆T_{vb}=\frac{\left(t_{vb,g}-t_{h,g}\right)}{ln\frac{\left(t_{vb,g}-t_{V}\right)}{\left(t_{h,g}-t_{V}\right)}}$$

The temperature, $t_{vb,g}$, on the heating-gas side results from the energy balance of the boiling zone as follows:(94)$$t_{vb,g}=t_{h,g}+\frac{{\dot{Q}}_{vb}}{{\dot{m}}_{g}\cdot c_{g}}$$

Overheating area

(95)$$A_{h}=\frac{{\dot{Q}}_{oh}}{U_{oh}\cdot ∆T_{m}^{oh}}=\frac{{\dot{m}}_{ORC}(t_{3}-t_{V})}{U_{oh}\cdot ∆T_{m}^{oh}}$$
where $U_{oh}=50\left[W/(m^{2}\cdot K)\right]$ is the global heat-transfer coefficient for the overheating process as follows:(96)$$∆T_{oh}=\frac{\left(t_{\alpha }-t_{3}\right)-\left(t_{vb,g}-t_{30}\right)}{ln\frac{\left(t_{\alpha }-t_{3}\right)}{\left(t_{vb,g}-t_{30}\right)}}$$
where $t_{\alpha }=420\;{}^{\circ}C$.

### 3.4. Exergoeconomic Performance Indicators

#### 3.4.1. Monetary Cost of Zonal Exergy Destruction

To calculate the monetary cost of the exergy destruction in each functional area, the value of the zonal exergy destruction is multiplied by the unitary monetary cost of the local zonal fuel (Table 5).

#### 3.4.2. Exergoeconomic Factor

To appreciate how the investment expense saves the monetary cost of the zonal exergy destruction, the exergoeconomic factor [31] is calculated for each piece of equipment, where the index, k, represents the piece of equipment as follows:(97)$$f_{k}=\frac{{\dot{Z}}_{k}}{{\dot{Z}}_{k}+{\dot{C}}_{I,k}}<1$$

If $f_{k}$ approaches 1, the operational cost (of exergy destruction, ${\dot{C}}_{I,k}$) is too low and the investment cost, ${\dot{Z}}_{k}$, is too high. In this case, the exergetic performance of the device must be relaxed, and a higher exergy destruction should be accepted; the total cost, consisting of the operating and investment costs, will decrease.

If $f_{k}$ is too low, the monetary value of the exergy destruction is too high, and the purchase of better equipment, which is obviously more expensive, is recommended.

#### 3.4.3. Zonal Coefficients of Performance

(a)Exergetic Efficiency

For each functional area, the product and fuel are specified (Table 5). Their ratio gives the zonal exergetic efficiency [31] as follows:(98)$${ɳ}_{ex,k}={\left(\frac{P}{F}\right)}_{k}100$$

For the global ORC-CVRC system, the product is the heat exergy extracted from the chilled water in the VCRC evaporator (evaporator product (Table 4)), and the fuel is the exergy of the combustion gases, ${\dot{E}x}_{\alpha }$; therefore, Equation (98) can be written as follows:(99)$${ɳ}_{ex}=\frac{P_{Ev}}{{\dot{E}x}_{\alpha }}100$$

(b)The relative weight of the exergy destruction in the fuel consumption of the global ORC-VCRC system

The weight percentage of the zonal exergy destruction in the fuel consumption of the global system is calculated using the following relationship:(100)$$\psi_{k}=\frac{I_{k}}{{\dot{E}x}_{\propto }}100$$

The search procedure for the optimal functional and constructive ORC-VCRC system will be guided by the values of the exergoeconomic performance indicators.

### 3.5. Results of Exergoeconomic Optimization

The following tables show the simulation results for the working fluid, R1224yd(Z), depending on the variation in several decision-making parameters, including the minimum temperature difference in the condenser ($∆T_{minCd}$), the minimum temperature difference in the evaporator (ΔT_minV_), the minimum temperature difference in the boiler (ΔT_minB_), as well as the variation in the efficiencies of the compressor (η_sCp_) or the expander (η_sE_). The unit cost of diesel was considered, $c_{Diesel}=1.7\;[EUR/l]$.

For ΔT_minCd_ = 8 K, ΔT_minV_ = 8 K, ΔT_minB_ = 30 K, η_sCp_ = 0.8, η_sP_ = 0.8 and η_sE_ = 0.8, the results of the simulation are given in Table 6.

For the above-mentioned decisional variables, the cost of one unit of exergy of the product, c_R_ [EUR/kJ] (refrigerating power of the VCRC evaporator) is as follows:$$c_{R}=13.15\cdot 10^{-4}\left[EUR/kJ\right]$$

Examining Table 6, one observes that the lowest exergetic efficiency, η_ex_, and the highest relative destruction, ψ, are in the boiler. The low exergoeconomic factor, f, requires the choice of a larger heat-exchange surface (a smaller minimum temperature difference) to reduce the amount of exergy destruction. 

Dropping the minimum temperature difference in the boiler to ${\Delta T}_{minB}=20\;K$, the unitary cost of the refrigerating power becomes, $c_{R}=12.35\cdot 10^{-4}\;\left[EUR/kJ\right]$, and further, for ${\Delta T}_{minB}=10\;K$, this cost decreases to $c_{R}=11.73\cdot 10^{-4}\;\left[EUR/kJ\right]$.

The same Table 6, shows that the VCRC evaporator has the lowest exergoeconomic factor, f = 0.002579, which indicates a high cost of exergy destruction compared to the cost for amortizing the invested capital. The suggestion is to reduce the temperature difference and, thus, increase the heat-exchange surface area of the evaporator. The minimum temperature difference in the evaporator was reduced to ΔT_minV_ = 3 K.

The results of the simulation for ΔT_minV_ = 3 K, are shown in Table 7.

After reducing the minimum temperature difference in the evaporator to ΔT_minV_ = 3 K, the cost of the product exergy unit became $c_{R}=9.681\cdot 10^{-4}\left[EUR/kJ\right]$.

Continuing the optimum search, one observes from Table 7, that now, the condenser has the lowest exergoeconomic factor, f = 0.00718. The conclusion is that the amount of exergy destruction, in the condenser, must be reduced.

Although the condenser is an eminently dissipative area with the role of transferring to the environment the energy generated and transferred in the system and, therefore, does not have an exergetic product, by specifying the condensation temperature, the condenser has an important role in establishing the operation efficiency.

In the aim of decreasing the exergy destruction in the condenser, the minimum temperature difference in this heat exchanger was reduced to ΔT_minCd_ = 3 K. The results of the simulation, in this last case, are presented in Table 8.

The cost of one product exergy unit becomes $c_{R}=7.585\cdot 10^{-4}\left[EUR/kJ\right]$.

The cost of the exergy destruction in the pump, as well as the cost for amortizing the purchase of the pump, are the lowest in the system (Table 8).

The highest value of the exergoeconomic factor of the pump area indicates the possibility for reducing the quality of the pump by reducing its isentropic efficiency, η_sP_; but the effect of this reduction on the thermodynamic parameters after pumping does not decrease the unit cost of the global-system product represented by the cold obtained in the evaporator.

Table 8 shows that the compressor has a low acquisition cost, $\dot{Z,}$ but induces a high cost of exergy destruction, $\dot{CI}$, which suggests the choice of a more expensive compressor with a higher isentropic compression efficiency, ${ɳ}_{sCp}.$ The results of the simulation when the isentropic efficiency of the compressor is increased to ${ɳ}_{sCp}=0.85$, are shown in Table 9.

The cost of the product exergy unit is $c_{R}=7.165\cdot 10^{-4}\left[EUR/kJ\right]$; and if one considers η_sCp_ = 0.89, the following results are obtained (Table 10).

For the decisional variables of Table 10, the cost of one unit of product exergy is $c_{R}=7.009\cdot 10^{-4}\left[EUR/kJ\right]$.

In the area of the expander, the cost of the exergy destruction is high, which indicates the possibility for reducing the exergy-destruction expense by choosing a more efficient expander. The results of the simulation when the isentropic efficiency is increased to η_sE_ = 0.85, are given in Table 11.

In this case, one unit of product exergy costs $c_{R}=6.632\cdot 10^{-4}\left[EUR/kJ\right]$.

With a higher-performance expander (η_sE_ = 0.9), one obtains the results listed in Table 12.

Under these conditions exergy cost of the product unit becomes $c_{R}=6.478\cdot 10^{-4}\left[EUR/kJ\right]$. The further increase in the isentropic efficiency of the expander is tempered by the expander’s cost which became higher than, for example the cost of the compressor (Table 12).

Following the exergoeconomic procedure (Table 6, Table 7, Table 8, Table 9, Table 10, Table 11 and Table 12), the optimal functional and constructive solution was obtained. The minimum cost of the cold unit, becomes $c_{R}=6.478\cdot 10^{-4}\left[EUR/kJ\right],$ is reached when the compound system ORC-VCRC operates with the decisional parameters considered in Table 12.

The cost of the cold unit decreased by half compared to the initial situation (Table 6) when $c_{R}=13.15\cdot 10^{-4}\left[EUR/kJ\right]$.

The impact of the fuel cost of the global ORC-VCRC system (diesel fuel) on the cost of the cold unit is presented in Figure 10.

## 4. Discussion

The presented study refers to the heat recovery of the combustion gases of a stationary internal combustion engine through the cogeneration of mechanical power and cold.

The heat recovery was considered as a theme of the project, for which the combustion gases were considered as an unused part of the fuel used in the engine to produce mechanical power and represent the fuel for the refrigeration plant driven by the ORC system.

If the project did not require the recovery of the energy of the combustion gases, this would be considered as an accepted loss, a situation in which the combustion gases are assigned a zero cost ($c_{gMAI}=0$).

It is interesting to note that in this case, contrary to the impression that if the cost of the fuel does not matter, the optimal economic solution is obtained for the cheapest and, therefore, non-performing equipment; however, the actual situation is not like that because the lower performance of the equipment affects the product of the system, i.e., the cold product. This is due to the conditions of this case, namely, that the potential of the recoverable energy from the combustion gases, although free, is limited. The solution to the optimization problem is to obtain the maximum amount of product (minimum unit cost of the product) from a limited resource at the minimum monetary cost.

To find the optimal operating and design conditions, the exergoeconomic analysis and optimization method were used, which is the only method that looks for the optimal solution, offering users clear rules for improving the studied system, for which the effects can be followed step by step.

Unlike exergoeconomic analysis, any other optimization procedure based on statistical, evolutionary, or mathematical algorithms represents black boxes for the user, and the results must be accepted on faith. In addition, any mathematical optimization method performs the search within the limits of the specified scheme without providing any hints on its structural improvement.

As proof of the power of exergy analysis to suggest structural changes in the system and reduce the amount of internal consumption of usable energy (reduction in the amount of exergy destruction), an exergy analysis is presented to highlight the high amount of exergy destruction in the ORC boiler and suggests that to reduce the amount exergy destruction, an internal heat exchanger should be used.

But the value of the exergy destruction quantified only from a thermodynamic point of view does not define the conditions where in addition to the interaction with the physical environment, the interaction of the system with its economic environment must also be considered. For this purpose, to assign an economic cost to each exergy destruction, a strategy was followed to find the unit cost of each substance and energy stream as a part of the cost of the resources purchased from outside the system and the cost for amortizing the equipment components.

The system was divided into functional areas for which the exergetic resource (generically, the fuel), product, and destruction were highlighted.

For each piece of equipment, an exergoeconomic correlation was built to provide a connection between the purchase cost, size of the exergetic product, and exergetic performance parameter.

Unlike thermoeconomic correlations, which calculate costs based on material consumption, exergoeconomic correlations provide an image of the sensitivity of the monetary cost for equipment depending on its exergetic performance (that is, the exergy destruction induced by the magnitude of the irreversibility of internal processes).

The modification of the exergetic and exergoeconomic performance coefficients to the changes made in the system guided the optimization procedure.

Owing to the high cost of the exergy destruction associated with the processes in the system, the exergoeconomic optimization procedure seeks to reduce them by increasing the cost of the investment in more efficiently performing equipment.

The cost of the fuel required for driving the thermal engine (the fuel of the global system) has a substantial influence on the cost of the cold unit.

## 5. Conclusions

The exergoeconomic optimization procedure takes into account the interaction of the system with its physical and economic environments, looking for the functional and constructive conditions for which the unit cost of the product is minimal. The total cost of the system over a period of time is taken into account and calculated as the sum of the operating cost and the amortization cost of the invested capital.

The analysis proposes that the optimal solution is to invest in larger or higher-performance equipment that will reduce the amount of exergy consumption in the system.

Despite increasing the investment cost with larger or higher-performance pieces of equipment, the higher rate of decrease in the operational cost leads to a reduction in the monetary cost of the final product toward the optimal constructive and functional solution.

When reducing the temperature differences in the system heat exchangers (ORC boiler, condenser, and VCRC evaporator), the unitary cost of the refrigeration drops by 44%. The increase in the isentropic efficiency of the ORC expander or VCRC compressor further reduces the unitary cost of refrigeration by another 15%.

As expected from the initial exergy analysis, the ORC boiler had increased influence for decreasing the amount of exergy destruction by increasing the heat-transfer surface area, which reduced the unitary cost of the final product by 26%, followed by the evaporator at 21% and the condenser at 6%. This makes sense because the lower the temperature level at which exergy is destroyed (consumed) the higher is its cost. The increase in the isentropic efficiency and cost of the expander and compressor is accompanied by a reduction in the unitary cost of the refrigeration (the final product of the combined system) by 8% for the expander and 2% for the compressor. This reduced contribution of the compressor and expander is due to their rapid purchase cost increase with the demand for higher isentropic efficiency. Although for an increase of 10% in the isentropic efficiency of the compressor, its purchase cost almost doubles, for a decrease of 66% in the temperature difference in the boiler, its purchase cost increases by only 7%.

Following the optimization procedure, the cost of the cooling unit drops by half. The cost of diesel fuel has a major influence on the unit cost of cooling. A doubling of the cost of diesel fuel leads to an 80% increase in the cost of the cold unit.

The exergoeconomic analysis is the only one that offers research engineers a methodology to search for the optimal conditions step by step and shows the immediate effects of the functional and performance changes in the equipment on the final product. In this way, the level of understanding of the processes that take place in the system, the connections between them, and the design of the equipment increases.

The originality of the proposed exergoeconomic optimization, compared to other mathematical approaches, consists of conducting the optimal constructive and parametric search in open view, providing permanent insights into the changes made to the system.

## Figures and Tables

**Figure 1 entropy-25-01531-f001:**
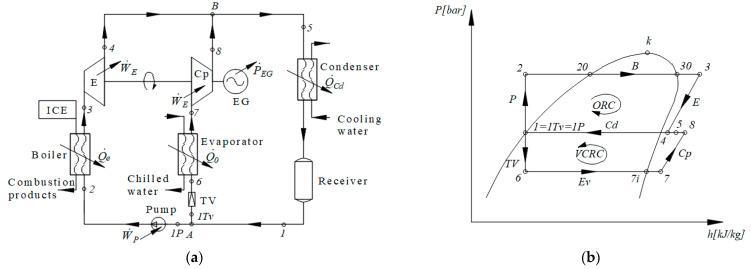
ORC-VCRC system: (**a**) flowchart; (**b**) ORC-VCRC cycles in the p–h diagram.

**Figure 2 entropy-25-01531-f002:**
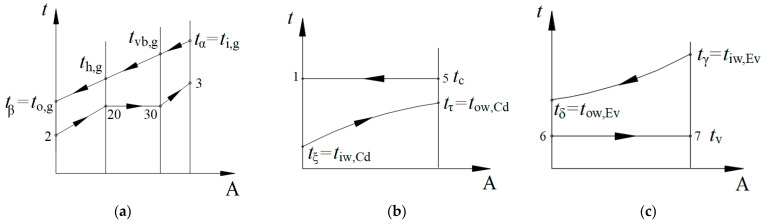
Temperature–Area diagrams: (**a**) ORC boiler; (**b**) ORC-VCRC condenser; (**c**) VCRC evaporator (for the ORC-VCRC, working fluid notations correspond to those in Figure 1).

**Figure 3 entropy-25-01531-f003:**
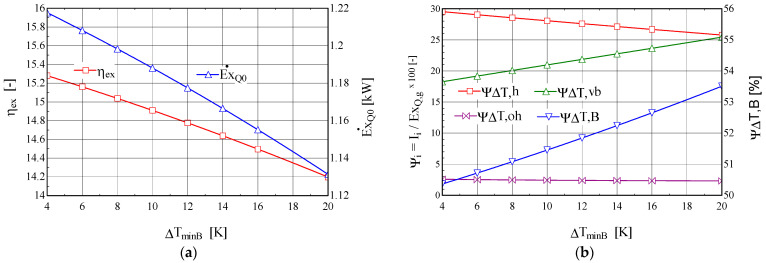
The ORC-VCRC system: (**a**) the exergetic performance coefficient and the exergy of the heat current extracted in the VCRC evaporator, depending on the minimum temperature difference, $∆t_{minB}$, in the ORC boiler. (**b**) The shares of the exergy consumption in the overall system, and the exergy destructions from the heating, boiling, and overheating areas of the boiler, depending on the minimum temperature difference, $∆t_{minB}$.

**Figure 4 entropy-25-01531-f004:**
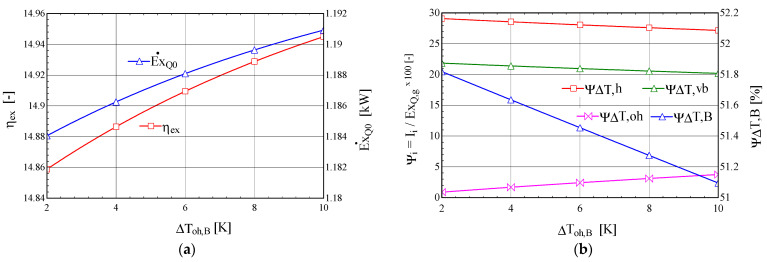
The ORC-VCRC system: (**a**) the exergetic performance coefficient and the exergy of the heat stream extracted in the VCRC evaporator, depending on the degree of overheating in the ORC boiler, $∆t_{oh,B}$. (**b**) The shares of the exergy consumption in the global system, and the exergy destructions from the heating, boiling, and overheating areas of the boiler, depending on the degree of overheating in the ORC boiler, $∆t_{oh,B}$.

**Figure 5 entropy-25-01531-f005:**
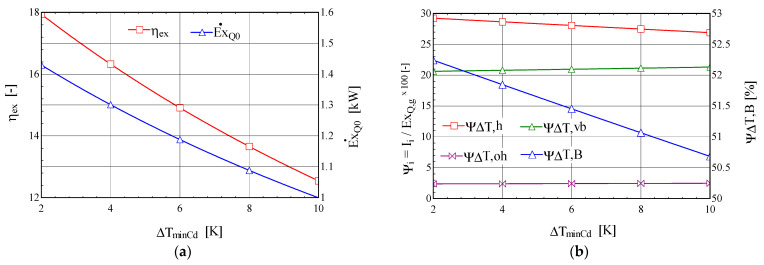
The ORC-VCRC system: (**a**) the exergetic performance coefficient and the exergy of the heat current extracted in the VCRC evaporator, depending on the temperature difference, $∆t_{minCD}$, in the condenser. (**b**) The shares of the exergy consumption in the overall system, and the exergy destructions in the heating, boiling, and overheating areas of the boiler, depending on the temperature difference, $∆t_{minCD}$, in the condenser.

**Figure 6 entropy-25-01531-f006:**
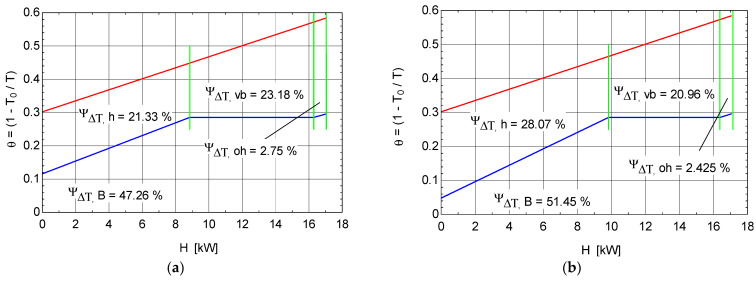
θ-H diagram of hot and cold currents from the ORC boiler: (**a**) simple ORC system (Figure 1); (**b**) ORC system with internal recuperative heat exchanger (Figure 7).

**Figure 7 entropy-25-01531-f007:**
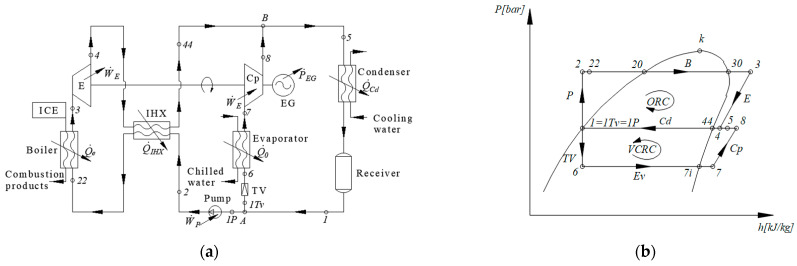
ORC-VCRC system containing ORC with internal regenerative heat exchanger; (**a**) scheme of the ORC-VCRC system; (**b**) the thermodynamic cycle of the ORC-VCRC system in the p–h diagram.

**Figure 9 entropy-25-01531-f009:**
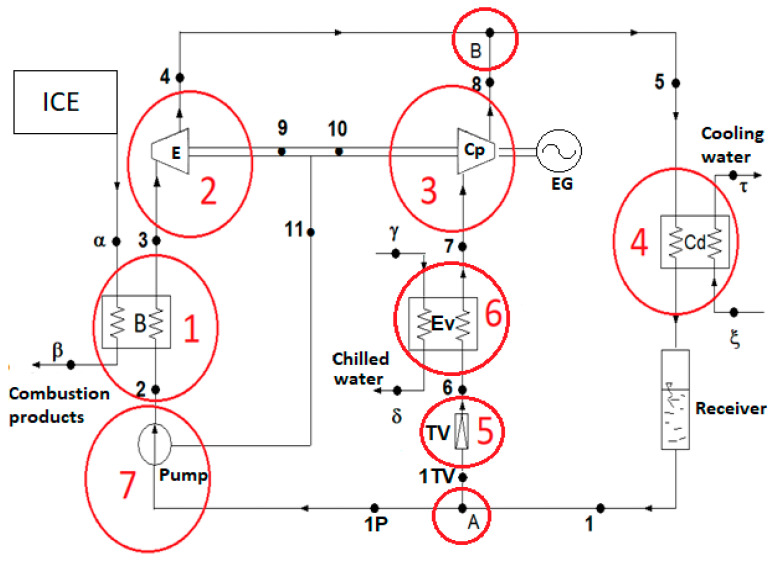
The ORC-VCRC system divided into functional areas.

**Figure 10 entropy-25-01531-f010:**
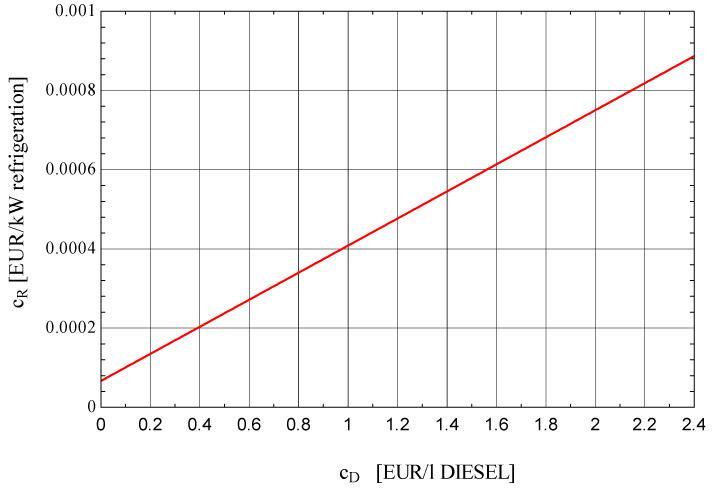
Influence of the diesel fuel unitary cost on the unitary cost of the refrigeration ΔT_minCd_ = 3 K; ΔT_minV_ = 3 K; ΔT_minB_ = 10 K; η_sCp_ = 0.89; η_sE_ = 0.9; η_sP_ = 0.8.

**Table 1 entropy-25-01531-t001:** The characteristics of the ICE-ORC-VCRC coupled system.

Main System Parameter	Value	Unit of Measure
Maximum mechanical power	40	kW
The temperature of the combustion gases in the 100% regime, equal to the temperature of the hot fluid at the entrance to the ORC boiler, ${t_{\propto }=t}_{i,g}$	420	°C
Outlet temperature of the combustion gases from the boiler, ${t_{\beta }=t}_{o,g}$	140	°C
Measured mass-flow rate of combustion gases, m_g_	0.0534	kg/s
Specific heat of combustion gases, $c_{g}$	1.14	kJ/kgK
The temperature of the cooling water at the entrance to the condenser, $t_{\xi }=t_{iw,Cd}$	15	°C
$\mathrm{Condenser}\;\mathrm{outlet}\;\mathrm{water}\;\mathrm{temperature},\;t_{\tau }=t_{ow,Cd}$	22	°C
Condenser cooling water mass-flow rate, ${\dot{m}}_{w,c}$	0.5	kg/s
The temperature of the cooled water at the entrance to the evaporator, $t_{\gamma }=t_{iw,Ev}$	8	°C
The temperature of the cooled water at the exit from the evaporator, ${t_{\delta }=t}_{ow,Ev}$	3	°C
Ambient temperature, $t_{o}$	15	°C
ORC-VCRC working fluid	R1224yd(Z)	-

**Table 2 entropy-25-01531-t002:** The results of the energetic and exergetic analyses for the basic ORC-VCRC scheme.

${COP}_{VCRC}$ **(-)**	${COP}_{ORC}$ **(-** **)**	${COP}_{ORC-VCRC}$ **(-** **)**	${ɳ}_{ex}$ **(%)**	${\dot{m}}_{VCRC}$ **(kg/s)**	${\dot{m}}_{ORC}$ **(kg/s)**	${\dot{m}}_{ORC-VCRC}$ **(kg/s)**	$\psi_{Cp}$ **(%)**	$\psi_{E}$ **(%)**	$\psi_{∆T,B}$ **(%)**	$\psi_{∆T,h}$ **(%)**	
6.108	0.1657	0.9468	14.91	0.1089	0.07605	0.1849	4.605	5.464	51.45	28.07	
$\psi_{∆T,oh}$ **(%)**	$\psi_{∆T,vb}$ **(%)**	$\psi_{P}$ **(%)**	$\psi_{L,Cd}$ **(%)**	$\psi_{TV}$ **(%)**	$\psi_{m}$ **(%)**	${\dot{Q}}_{B}$ **(kW)**	${\dot{Q}}_{Cd}$ **(kW)**	${\dot{Q}}_{Ev}$ **(kW)**	${\dot{W}}_{E}$ **(kW)**	${\left|\dot{W}\right|}_{P}$ **(kW)**	${\left|\dot{W}\right|}_{Cp}$ **(kW)**
2.425	20.96	0.8721	19.22	3.21	0.267	17.05	33.18	16.14	2.825	0.1823	2.6427

**Table 3 entropy-25-01531-t003:** Energy and exergy results obtained for the ORC-VCRC scheme with internal heat exchanger on the ORC circuit.

${COP}_{VCRC}$ **(-)**	${COP}_{ORC}$ **(-)**	${COP}_{ORC-VCRC}$ **(-)**	${ɳ}_{ex}$ **(%)**	${\dot{m}}_{VCRC}$ **(kg/s)**	${\dot{m}}_{ORC}$ **(kg/s)**	${\dot{m}}_{ORC-VCRC}$ **(kg/s)**	$\psi_{Cp}$ **(%)**	$\psi_{E}$ **(%)**	$\psi_{∆T,B}$ **(%)**	$\psi_{∆T,h}$ **(%)**	
6.108	0.1883	1.076	16.94	0.1237	0.08642	0.2101	5.233	6.21	47.26	21.33	
$\psi_{∆T,oh}$ **(%)**	$\psi_{∆T,vb}$ **(%)**	$\psi_{P}$ **(%)**	$\psi_{L,Cd}$ **(%)**	$\psi_{TV}$ **(%)**	$\psi_{m}$ **(%)**	${\dot{Q}}_{B}$ **(kW)**	${\dot{Q}}_{Cd}$ **(kW)**	${\dot{Q}}_{Ev}$ **(kW)**	${\dot{W}}_{E}$ **(kW)**	${\left|\dot{W}\right|}_{P}$ **(kW)**	${\left|\dot{W}\right|}_{Cp}$ **(kW)**
2.751	23.18	0.991	19.34	3.648	0.1194	17.05	33.38	18.34	3.21	0.2072	3.0028

**Table 4 entropy-25-01531-t004:** The concepts of Fuel, Product, Exergy Loss, and Destruction for functional areas.

Equipment	Fuel	Product	Exergy Loss	Destruction
Boiler	Ėxα	${Ėx}_{3}-{Ėx}_{2}$	Ėxβ	${Ėx}_{\alpha }-{Ėx}_{3}+{Ėx}_{2}-{Ėx}_{\beta }$
Expander	${Ėx}_{3}-{Ėx}_{4}$	$\left|\dot{W_{D}}\right|$		${Ėx}_{3}-{Ėx}_{4}-{\dot{W}}_{D}$
Compressor	$\left|\dot{W_{Cp}}\right|$	${Ėx}_{8}-{Ėx}_{7}$		$\left|\dot{W_{Cp}}\right|-{Ėx}_{8}+{Ėx}_{7}$
Condenser	${Ėx}_{5}-{Ėx}_{1}$		${Ėx}_{\tau }-{Ėx}_{\xi }$	${Ėx}_{5}-{Ėx}_{1}-{Ėx}_{\tau }+{Ėx}_{\xi }$
Throttling Valve	${\dot{E}x}_{1TV}$	Ėx_6_		${Ėx}_{1TV}-{Ėx}_{6}$
Evaporator	${Ėx}_{6}-{Ėx}_{7}$	${Ėx}_{\delta }-{Ėx}_{\gamma }$		${Ėx}_{6}-{Ėx}_{7}-{Ėx}_{\delta }+{Ėx}_{\gamma }$
ORC Pump	$\left|\dot{W_{P}}\right|$	${Ėx}_{2}-{Ėx}_{1P}$		$\left|\dot{W_{P}}\right|-{Ėx}_{2}+{Ėx}_{1P}$

**Table 5 entropy-25-01531-t005:** Unit cost of zonal fuel and monetary cost rate of zonal exergy destruction.

Equipment	$\mathrm{Zonal}\;\mathrm{Fuel}\;\mathrm{Unit}\;\mathrm{Cost}\;\left[\frac{\mathrm{EUR}}{kJex}\right]$	$\mathrm{Monetary}\;\mathrm{Cost}\;\mathrm{Rate}\;\mathrm{of}\;\mathrm{Zonal}\;\mathrm{Exergy}\;\mathrm{Destruction},\;{\dot{C}}_{I}$ $\left[\frac{EUR}{s}\right]$
Boiler	$c_{gMAI}$	$c_{gMAI}\left({\dot{Ex}}_{\alpha }-{\dot{Ex}}_{3}+{\dot{Ex}}_{2}-{\dot{Ex}}_{\beta }\right)$
Expander	$c_{3}$	$c_{3}\left({\dot{Ex}}_{3}-{\dot{Ex}}_{4}-{\dot{W}}_{D}\right)$
Compressor	$c_{10}$	$c_{10}\left(\left|{\dot{W}}_{Cp}\right|-{\dot{Ex}}_{8}-{\dot{Ex}}_{7}\right)$
Condenser	$c_{5}$	$c_{5}\left({\dot{Ex}}_{5}-{\dot{Ex}}_{1}-{\dot{Ex}}_{\tau }+{\dot{Ex}}_{\xi }\right)$
Throttling Valve	$c_{1}$	$c_{1}\left({\dot{Ex}}_{A}-{\dot{Ex}}_{6}\right)$
Evaporator	$c_{6}$	$c_{6}\left({\dot{Ex}}_{6}-{\dot{Ex}}_{7}-{\dot{Ex}}_{\delta }+{\dot{Ex}}_{\gamma }\right)$
ORC Pump	$c_{11}$	$c_{11}\left(\left|{\dot{W}}_{P}\right|+{\dot{Ex}}_{1p}-{\dot{Ex}}_{2}\right)$

**Table 6 entropy-25-01531-t006:** The results obtained for ΔT_minCd_ = 8 K; ΔT_minV_ = 8 K; ΔT_minB_ = 30 K; η_sCp_ = 0.8; η_sP_ = 0.8; η_sE_ = 0.8.

Zone	ηex (%)	$\dot{I}$ (kW)	Ψ (%)	Ż (EUR/h)	CI (EUR/kJ)	$\dot{C}I=CI\cdot \dot{I}$ (EUR/h)	ĊI + Ż (EUR/h)	f
Boiler	31.43	4.211	37.17	0.01067	4.07·10^−5^	0.617	0.6277	0.01701
Expander	82.25	0.4939	4.359	0.01204	2.021·10^−4^	0.3593	0.3713	0.03242
Compressor	81.65	0.4035	3.561	0.005774	2.471·10^−4^	0.359	0.3648	0.01583
Condenser	-	1.2	13.63	0.00695	2.535·10^−4^	1.095	1.102	0.006306
VCRC Evaporator	41.37	0.5094	4.496	0.002576	5.432·10^−4^	0.9962	0.9988	0.002579
ORC Pump	81.03	0.01719	0.1517	0.002539	2.471·10^−4^	0.0153	0.01783	0.1423

**Table 7 entropy-25-01531-t007:** The results obtained for ΔT_minCd_ = 8 K; ΔT_minB_ = 10 K; η_sCp_ = 0.8; η_sP_ = 0.8; η_sE_ = 0.8 when ΔT_minV_ = 3 K is imposed.

Zone	ηex (%)	I (kW)	Ψ (%)	Ż (EUR/h)	CI (EUR/kJ)	$\dot{C}I=CI\cdot \dot{I}$ (EUR/h)	ĊI + Ż (EUR/h)	f
Boiler	34.59	3.852	34	0.01137	4.07·10^−5^	0.5645	0.5758	0.01975
Expander	82.43	0.5559	4.906	0.01372	1.723·10^−4^	0.3448	0.3585	0.03826
Compressor	81.61	0.4547	4.013	0.005171	2.105·10^−4^	0.3446	0.3498	0.01479
Condenser	-	1.376	15.64	0.007987	2.228·10^−4^	1.103	1.111	0.00718
VCRC Evaporator	56.41	0.3789	3.344	0.005873	5.443·10^−4^	0.7425	0.7484	0.00784
ORC Pump	81.05	0.02545	0.2246	0.004271	2.105·10^−4^	0.01928	0.02356	0.1813

**Table 8 entropy-25-01531-t008:** The results obtained for ΔT_minV_ = 3 K; ΔT_minB_ = 10 K; η_sCp_ = 0.8; η_sP_ = 0.8; η_sE_ = 0.8 when ΔT_minCd_ = 3 K is imposed.

Zone	ηex (%)	I (kW)	Ψ (%)	Ż (EUR/h)	CI (EUR/kJ)	$\dot{C}I=CI\cdot \dot{I}$ (EUR/h)	ĊI + Ż (EUR/h)	f
Boiler	33.91	3.93	34.68	0.01117	4.07·10^−5^	0.5758	0.5869	0.01904
Expander	82.26	0.5872	5.182	0.01432	1.615·10^−4^	0.3415	0.3558	0.04024
Compressor	81.33	0.4837	4.269	0.004574	1.979·10^−4^	0.3445	0.3491	0.0131
Condenser	-	0.9541	12.4	0.01742	2.147·10^−4^	0.7374	0.7548	0.02307
VCRC Evaporator	56.38	0.4878	4.305	0.007549	4.258·10^−4^	0.7476	0.7552	0.009996
ORC Pump	80.73	0.02535	0.2237	0.0053	1.979·10^−4^	0.01805	0.02335	0.2269

**Table 9 entropy-25-01531-t009:** The results obtained for ΔT_minCd_ = 3 K; ΔT_minV_ = 3 K; ΔT_minB_ = 10 K; η_sP_ = 0.8; η_sE_ = 0.8 when η_sCp_ = 0.85 is imposed.

Zone	ηex (%)	I (kW)	Ψ (%)	Ż (EUR/h)	CI (EUR/kJ)	$\dot{C}I=CI\cdot \dot{I}$ (EUR/h)	ĊI + Ż (EUR/h)	f
Boiler	33.91	3.93	34.68	0.01117	4.07·10^−5^	0.5758	0.5869	0.01904
Expander	82.26	0.5872	8.182	0.01432	1.583·10^−4^	0.3347	0.349	0.04102
Compressor	85.97	0.3636	3.209	0.009721	1.939·10^−4^	0.2538	0.2635	0.03688
Condenser	-	0.972	12.7	0.01801	2.04·10^−4^	0.7139	0.7319	0.02461
VCRC Evaporator	56.38	0.5183	4.574	0.008021	4.021·10^−4^	0.7501	0.7582	0.01058
ORC Pump	80.73	0.02535	0.2237	0.0053	1.939·10^−4^	0.0177	0.023	0.2305

**Table 10 entropy-25-01531-t010:** The results obtained for ΔT_minCd_ = 3 K; ΔT_minV_ = 3 K; ΔT_minB_ = 10 K; η_sP_ = 0.8; η_sE_ = 0.8 when η_sCp_ = 0.89 is imposed.

Zone	ηex (%)	I (kW)	Ψ (%)	Ż (EUR/h)	CI (EUR/kJ)	$\dot{C}I=CI\cdot \dot{I}$ (EUR/h)	ĊI + Ż (EUR/h)	f
Boiler	33.91	3.93	34.68	0.01117	4.07·10^−5^	0.5758	0.5869	0.01904
Expander	82.26	0.5872	5.182	0.01432	1.571·10^−4^	0.3322	0.3465	0.04132
Compressor	89.69	0.2761	2.357	0.05089	1.925·10^−4^	0.185	0.2359	0.2157
Condenser	-	0.9868	12.94	0.01849	2.005·10^−4^	0.7123	0.7308	0.0253
VCRC Evaporator	56.38	0.5426	4.789	0.008398	3.933·10^−4^	0.7683	0.7767	0.01081
ORC Pump	80.73	0.02535	0.2237	0.0053	1.925·10^−4^	0.01756	0.02286	0.2318

**Table 11 entropy-25-01531-t011:** The results obtained for ΔT_minCd_ = 3 K; ΔT_minV_ = 3 K; ΔT_minB_ = 10 K; η_sCp_ = 0.89; η_sP_ = 0.8 when η_sE_ = 0.85 is imposed.

Zone	ηex (%)	I (kW)	Ψ (%)	Ż (EUR/h)	CI (EUR/kJ)	$\dot{C}I=CI\cdot \dot{I}$ (EUR/h)	ĊI + Ż (EUR/h)	f
Boiler	33.91	3.93	34.68	0.01117	4.07·10^−5^	0.5758	0.5869	0.01904
Expander	86.74	0.4422	3.903	0.02454	1.54·10^−4^	0.2451	0.2697	0.09101
Compressor	89.69	0.2846	2.512	0.05423	1.799·10^−4^	0.1843	0.2385	0.2274
Condenser		1.009	13.3	0.01919	1.908·10^−4^	0.6934	0.7125	0.02693
VCRC Evaporator	56.38	0.5783	5.104	0.008949	3.72·10^−4^	0.7745	0.7835	0.01142
ORC Pump	80.73	0.02535	0.2237	0.0053	1.799·10^−4^	0.01641	0.02171	0.2441

**Table 12 entropy-25-01531-t012:** The results obtained for ΔT_minCd_ = 3 K; ΔT_minV_ = 3 K; ΔT_minB_ = 10 K; η_sCp_ = 0.89; η_sP_ = 0.8 when η_sE_ = 0.9 is imposed.

Zone	ηex (%)	I (kW)	Ψ (%)	Ż (EUR/h)	CI (EUR/kJ)	$\dot{C}I=CI\cdot \dot{I}$ (EUR/h)	ĊI + Ż (EUR/h)	f
Boiler	33.91	3.93	34.68	0.01117	4.07·10^−5^	0.5758	0.5869	0.01904
Expander	91.19	0.2961	2.613	0.0859	1.527·10^−4^	0.1627	0.2486	0.3455
Compressor	89.69	0.3021	2.667	0.05757	1.753·10^−4^	0.1906	0.2482	0.232
Condenser	-	1.033	13.66	0.01989	1.874·10^−4^	0.6967	0.7166	0.02775
VCRC Evaporator	56.38	0.3109	5.418	0.009501	3.634·10^−4^	0.8031	0.8126	0.01169
ORC Pump	80.73	0.02535	0.2237	0.0053	1.753·10^−4^	0.01599	0.02129	0.2489

## Data Availability

The study did not report any data.

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
