# Peer review of "Exergoeconomic Analysis of a Mechanical Compression Refrigeration Unit Run by an ORC"

_entropy, 2023, doi:10.3390/e25111531_

Round 1

Reviewer 1 Report

Comments and Suggestions for Authors

This is very interesting paper. Methodology is comprehensive and detailed and results are clearly presented and appropriately analysed.

My only concern is about the references. There are other papers that you could cite, especially those that consider the same working fluid as your paper.

Also, grouped references should be removed. For example, lines 81-82: "all thermal energy can be converted into 81 power and used for other applications [33-40]". There is no point citing paper in this way; it doesn't ad anything to the paper. Please review and amend.

Author Response

Dear Reviewer,

Thank you for the nice words of appreciation!

We are very grateful for the comments and observations that help us to improve our paper. Thank you!

Observation

My only concern is about the references. There are other papers that you could cite, especially those that consider the same working fluid as your paper.

Also, grouped references should be removed. For example, lines 81-82: "all thermal energy can be converted into 81 power and used for other applications [33-40]". There is no point citing paper in this way; it doesn't ad anything to the paper. Please review and amend.

Answer

The introduction has been reconsidered by adding 8 new references (16-23) dealing with close subjects. The references have been critically analyzed one by one. Groups of cited references have been omitted.

Thank you!

Reviewer 2 Report

Comments and Suggestions for Authors

See attached file for comments.

Comments on the Quality of English Language

Author Response

Dear Reviewer,

We are very grateful for the comments and observations that help us to improve our paper. Thank you!

ABSTRACT

Observation 1_ Stating the efficiency can be improved by “cold” is ambiguous. If waste heat recovery is the factor for “producing the cold” it should be emphasized, then cold does not need mentioning as it is an output.

Answer: The abstract was reformulated, and ambiguous stating were omitted

Observation 2_ More personal preference than comment, why is Mechanical Vapor Compression abbreviate to MRC instead of MVC?

Answer: Abbreviation MRC (chosen initially for its simplicity to make difference between mechanical and thermo-chemical compression) has been replaced by the more common VCRC (vapor compression refrigerating cycle)

Obsrevation 3_ See comment in conclusions. This is an over simplification of results and does not show the extent of the findings

Answer: The abstract has been completed with more detailed information about results and the original contribution

INTRODUCTION

Observation_1 First paragraph

  • The main purpose of a thermal system is storage/transfer/conversion of heat/energy, what was is also doing?
  • Perhaps provide more detail and explanation of relevant sources using not just low grade, but medium to high grade applications as well.
  • Needs re-wording, this sentence just defines what an ORC already achieves, not how other systems were combined with it or how they made it better.

Answer: The introduction has been mostly reformulated, and we considered all the above observations.

Observation 2_ about groups of cited works_ Needs expanding to include what these papers found and how it is relevant.

Answer: The introduction has been reconsidered by adding 8 new references (16-23) dealing with close subjects. The references have been critically analyzed one by one. Groups of cited references have been omitted.

Observation 3: After the state of art (line 140 in the first form of the paper)_A good selection of research is discussed, there needs to be more evaluation of findings and how that specifically supports the need for the research presented in this paper.

Answer: To clarify about our findings in the literature review that made us decide to bring our contribution, we added the following:

The literature review reveals the interest in the optimization of the compound system ORC-VCRC. In most approaches exergy analysis is used for estimating the magnitude of inefficiencies, and for the optimization procedure researchers appeal to mathematical methods.

In the present work the optimization of an ORC-VCRC will be conducted in an open view, based only on exergoeconomic principles. The exergoeconomic optimization offers the possibility to follow the result of any structural or operational local change upon the monetary cost of the overall system desired product.

Observation 4_ Table 1_ The parameters column needs to be presented clearer. Additionally, what determined the specific value of these parameters e.g. why 0.0534kg/s for the mass flow rate?

Answer: In table 1 we added a column with the unit measures.

The parameters in table 1 were experimentally measured on an ICE-ORC experimental stand.

3.3 Exergoeconomic cost correlations of component equipment

Observation 5_line 643_about the heat transfer in the boiler between the combustion gases and the ORC fluid _ why ?â„Ž = 70 [?/(?2 ?)]

Answer: the value of the overall heat transfer coefficient U is smaller than the smallest convection heat transfer coefficient between the two fluids exchanging heat. As the combustion gases are characterized by a small convection heat transfer coefficient (less than 100 W/m2K) we considered this value for U_h.

CONCLUSIONS

Observation: The conclusions currently feel like an over simplification of the results presented and need to better summarize all the findings from cost to optimal parameters assessed. This should then be reflected in the Abstract.

Answer: The conclusions have been reformulated and we added comments about the results obtained after the optimization procedure.

A resume of these comments is added in the ABSTRACT.

Thank you!

Reviewer 3 Report

Comments and Suggestions for Authors

The authors presented an exergo-econominc study of ORC-MCR system to investigate the potential of the capturing the waste heat in diesel engines. Please see comments below:

1- The abstract is the gate to your manuscript. It should contain answers to the following questions: What problem was studied and why is it important? What methods were used? What are the important results? What conclusions can be drawn from the results? What is the novelty of the work and where does it go beyond previous efforts in the literature?

2- The manuscript should be revised carefully in terms of English as there are some typo mistakes. In fact, you may need someone who is expert in the field of research to improve the writing. 

3- Page 1' line 42, ''thermal efficiency of the ORC system is quite low'' please elaborate.

4- page 2, line 49 and 51, avoid lumping the references, [9-11], [12-14]. Write a sentence or two about the findings of each reference.

5- ''Paper [15]''. mentioned the authors' names instead. 

6- Figure 1 and Figure 2 contain some scratches. Improve.

7- Add a paragraph to describe the operating principles of the system in Figure 1.

8- Does the chilled water in evaporator in Figure 1 cool or heat the cycle's working fluid?

9- page 5, lines 193 and 194, ''processes 2-20 or 2 - 2o??'' is it 0 or o?

10- Upon adding an internal heat exchager, the ORC mass changed from 0.07605 to 0.08642 kg/s. Explain.

11- Add the values of the ORC net work in Tables 2 and 3.

Comments on the Quality of English Language

The manuscript should be revised carefully in terms of English as there are some typo mistakes. In fact, the authors may need someone who is expert in the field of research to improve the writing. 

Author Response

Dear Reviewer,

We are very grateful for the comments and observations that help us to improve our paper. Thank you!

Observation 1- The abstract is the gate to your manuscript. It should contain answers to the following questions: What problem was studied and why is it important? What methods were used? What are the important results? What conclusions can be drawn from the results? What is the novelty of the work and where does it go beyond previous efforts in the literature?

Answer: We reformulated the Abstract trying to fulfill all the above mentioned requirements

Observation 2- The manuscript should be revised carefully in terms of English as there are some typo mistakes. In fact, you may need someone who is expert in the field of research to improve the writing. 

Answer: We did our best and carefully revised the English formulating and writing.

Observation 3-Page 1' line 42, ''thermal efficiency of the ORC system is quite low'' please elaborate.

Answer: Thank you – you are right – it is not an appropriate formulation – the Rankine cycle has good efficiency. We omitted this statement and reformulated.

Observation 4- page 2, line 49 and 51, avoid lumping the references, [9-11], [12-14]. Write a sentence or two about the findings of each reference

Answer: The introduction has been reconsidered by adding 8 new references (16-23) dealing with close subjects. The references have been critically analyzed one by one. Groups of cited references have been omitted.

Observation 5- ''Paper [15]''. mentioned the authors' names instead. 

Answer: We did mention that.

Observation 6- Figure 1 and Figure 2 contain some scratches. Improve.

Answer: We eliminated scratches and improved the figures

Observation 7- Add a paragraph to describe the operating principles of the system in Figure 1.

Answer: We added a paragraph bellow figure 1 with the description of the operating principles

Observation 8- Does the chilled water in evaporator in Figure 1 cool or heat the cycle's working fluid?

Answer: the chilled water transfers heat to the cycle’s working fluid. It is cooled down by the VCRC fluid.

Observation 9- page 5, lines 193 and 194, ''processes 2-20 or 2 - 2o??'' is it 0 or o?

Answer: thank you for the observation – it is 20 and 30 and we did corrections on the schematics

Observation 10- Upon adding an internal heat exchanger, the ORC mass changed from 0.07605 to 0.08642 kg/s. Explain.

Answer: The heating load Q_B of the ORC boiler is fixed (Eq. (4)). By adding an internal HX, h_2 increases (becoming h_22 (Figure 7)) while h_3 remains constant. The specific heating load of the boiler q_B=(h_3-h_22) [kJ/kg] decreases leading to an increase in the ORC mass flow rate (Eq. 5)

Observation 11- Add the values of the ORC net work in Tables 2 and 3.

Answer: We added these required values in Table 2 and 3. The net work of the ORC is the compressor mechanical work input W_cp

Thank you!

Round 2

Reviewer 3 Report

Comments and Suggestions for Authors

Authors have addresses all the comments.